# When Generalized Zero-Shot Learning Meets PU Learning: A Plug-and-Play Framework for Seen-Class Bias Mitigation

Long Tang [* 1]   Keyang Pu [* 1]   Yingjie Tian [2]

## Abstract

Generalized Zero-Shot Learning (GZSL) suffers from severe seen-class bias, a challenge stemming from the label incompleteness inherent in the mixed seen–unseen test set. To address this, we propose PUFE, a unified plug-and-play post-processing framework that recasts GZSL inference as a Positive-Unlabeled (PU) learning task by treating seen categories as positive and mixed test data as unlabeled. PUFE constructs a PU classifier in the semantic space, jointly estimating the seen-class posterior and labeling propensity via Maximum Likelihood Estimation (MLE) within a dual-head network. Furthermore, we introduce an adaptive prototype calibration strategy that employs high-confidence pseudo-instances—identified by the PU classifier—to explicitly align semantic prototypes with the underlying test distribution. Extensive experiments demonstrate that PUFE mitigates bias and raises the performance of diverse GZSL baselines across multiple architectural paradigms, yielding gains of up to 11.2 percentage points in the harmonic mean.

## 1. Introduction

Zero-Shot Learning (ZSL) aims to recognize novel categories without relying on any labeled samples from unseen classes by exploiting shared semantic knowledge across categories. In recent years, ZSL has increasingly embraced Transformer-based architectures and powerful pre-trained vision-language models, pushing the frontier of fine-grained recognition, cross-modal understanding, and open-world visual perception. As a pivotal extension of ZSL, Generalized

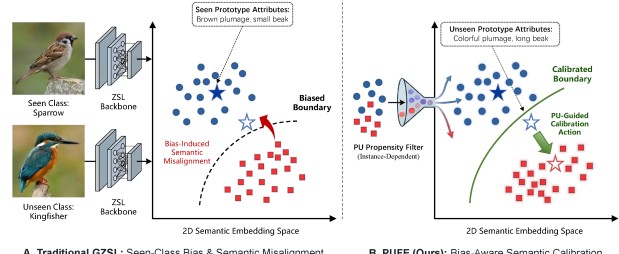

**Figure 1.** The mechanism of mitigating Seen-Class Bias: Baseline GZSL vs. PUFE-GZSL

ZSL (GZSL) aims to recognize instances from both seen and unseen categories without access to labeled unseen-class samples during training (Pourpanah et al., 2023). However, the absence of unseen-class data during training induces a pronounced **domain shift**, leading to severe bias towards seen categories in the test phase, where unseen instances are frequently misclassified as seen categories within the mixed label space (Chao et al., 2016).

Prevailing GZSL methods respond to this challenge predominantly by operating at the representation level, focusing on either strengthening visual-semantic alignment or compensating for absent unseen-class data. Representative approaches enhance embedding transferability via attention-based interaction or mutual distillation (Chen et al., 2022b;c; Liu et al., 2023; Chen et al., 2024c), while generative methods synthesize unseen-class features with GANs or VAEs to bridge distribution gaps (Xian et al., 2018b; Schonfeld et al., 2019; Chen et al., 2021a). More recently, state-space models (Hou et al., 2025) and vision-language pre-training (Chen et al., 2024a) have further broadened the architectural repertoire in GZSL. Despite their empirical success, these strategies remain indirect—refining representations or augmenting training features—without explicitly modeling the **label incompleteness** inherent in the mixed seen–unseen label space (Zhang & Li, 2025). Transductive GZSL partially alleviates this issue by leveraging unlabeled instances of unseen categories to adapt to the mixed test distribution (Cheng et al., 2025; Liu et al., 2024); yet access to a pre-separated unseen-class pool prior to inference is often impractical, substantially limiting real-world applicability.

[1] School of Artificial Intelligence, Nanjing University of Information Science & Technology, Nanjing 210044, China [2] Research Center on Fictitious Economy and Data Science, Chinese Academy of Sciences, Beijing 100190, China. Correspondence to: Long Tang <longt@nuist.edu.cn>.

*Proceedings of the 43rd International Conference on Machine Learning*, Seoul, South Korea. PMLR 306, 2026. Copyright 2026 by the author(s).

Conceptually, GZSL constitutes a label-incomplete learning problem: during training, only seen-class samples are labeled, whereas test samples are drawn from a mixture of both seen and unseen categories. Motivated by the structural correspondence, we cast "belonging to a seen category" as the positive state, thereby situating the training-test scenario within the Positive–Unlabeled (PU) setting (Bekker & Davis, 2020): seen-class training samples naturally constitute the positives, while the mixed test set remains entirely unlabeled. The perspective reframes bias mitigation as a structured test-time adaptation task: the naturally mixed test set—where class membership is unknown—provides exploitable structure for disentangling unseen instances via PU learning, without relying on any pre-isolated unseen subset.

Building upon these insights, we introduce **PUFE**, a **plug-and-play unified framework** serving as a seamless post-processing module for **embedding-based** GZSL. PUFE treats the backbone merely as a semantic feature generator and conducts PU learning directly within the semantic space, thereby preserving the backbone's inherent generalization capacity. Specifically, we devise a dual-head network to instantiate the PU classifier, which jointly estimates the instance-dependent seen-class posterior and labeling propensity via Maximum Likelihood Estimation (MLE). Guided by the established PU classifier, the resulting high-confidence pseudo-instances serve to calibrate their semantic prototypes. Such recalibration pulls prototypes back toward their true semantic neighborhoods, mitigating the seen-class bias.

The main contributions of this work can be summarized in the following three aspects.

(1) We establish a principled PU-learning perspective on GZSL inference and instantiate it as PUFE, a generic plug-and-play post-processing framework. This formulation achieves effective bias mitigation without prior class separation or backbone retraining, fitting naturally into a test-time transductive setting.

(2) We design a dual-head PU classifier to accurately estimate the seen-class posterior and labeling propensity from mixed test data. Moreover, we introduce a prototype calibration strategy that utilizes high-confidence pseudo-instances to explicitly align both seen and unseen semantic prototypes with the true data distribution at test time.

(3) Comprehensive evaluations on standard CUB, SUN, and AWA2 benchmarks across seven representative GZSL baselines demonstrate performance gains regardless of backbone architecture—with new state-of-the-art results established on CUB and AWA2— confirming PUFE's robustness in calibrating semantic prototypes across diverse architectural paradigms, while highlighting the promise of PU-based test-time adaptation for mitigating seen-class bias.

## 2. Related Work

### 2.1. Generalized Zero-Shot Learning

Generalized Zero-Shot Learning (GZSL) extends the scope of ZSL to realistic scenarios, requiring models to distinguish both seen and unseen classes at test time, where the two are intermixed (Pourpanah et al., 2023). Existing approaches are broadly classified into three representative paradigms.

**Embedding-based Methods.** Embedding-based GZSL learns compatibility mappings between visual features and semantic representations (Wang et al., 2019). Recent studies enhance cross-modal interaction and embedding discriminability via Transformer-based architectures or knowledge distillation, as exemplified by TransZero (Chen et al., 2022b) and PSVMA (Liu et al., 2023). These methods substantially improve representation quality and form the dominant backbone paradigm in contemporary GZSL research.

**Generative Methods.** Another line of work addresses unseen-class data scarcity by synthesizing visual features with generative models such as GANs or VAEs, effectively reducing GZSL to supervised learning (Xian et al., 2018b; Schonfeld et al., 2019; Chen et al., 2021a). Related prototype adaptation approaches further refine semantic centers to better align with visual feature distributions (Chen et al., 2023a; Hou et al., 2024).

**Calibration and Transductive Methods.** Bias mitigation strategies rebalance seen and unseen predictions via posterior calibration (Liu et al., 2018; Lu et al., 2024), while transductive GZSL draws on unlabeled unseen-class samples through distribution alignment, cluster-structure exploitation (Yue et al., 2024; Zhang et al., 2022), or iterative pseudo-label refinement with hardness sampling (Bo et al., 2021; Liu et al., 2024; Wang et al., 2023; Yang et al., 2024; Li et al., 2025).

In contrast to conventional transductive GZSL methods that presuppose a pre-separated unseen-class pool, our framework adopts a test-time transductive perspective and operates directly on the naturally mixed test data without any prior class-membership information, better reflecting practical deployment conditions. Our setting relates to test-time adaptation (TTA) (Wang et al., 2020; Sun et al., 2020), but differs in confronting label-space incompleteness rather than the covariate shift under a fixed label space assumed by general TTA.

### 2.2. Positive-Unlabeled Learning

As a core paradigm in weakly supervised learning (Zhou, 2018), PU learning addresses binary classification settings in which only positive samples are labeled, while unlabeled data may contain both positive and negative instances.

Early PU methods commonly adopt the Selected Completely

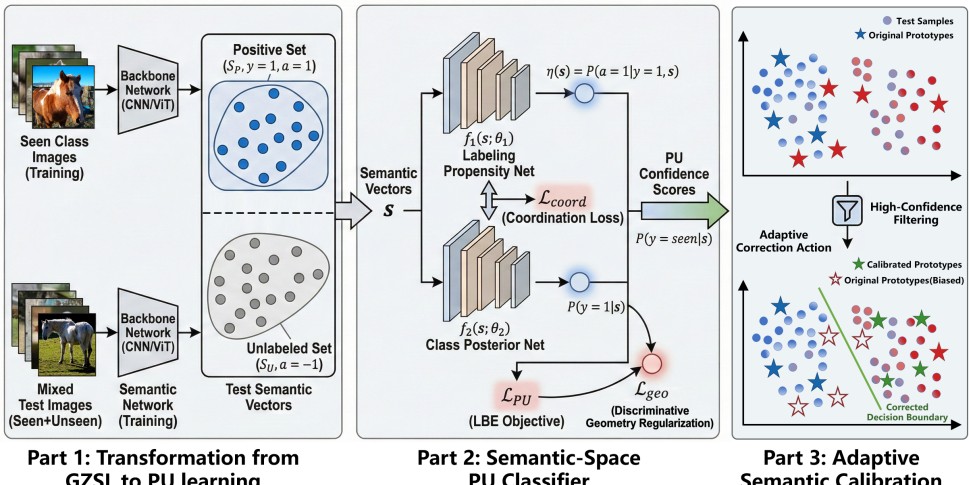

*Figure 2.* **Overview of the PUFE framework.** PUFE treats seen classes as positives and the mixed test set as unlabeled in semantic space, and learns labeling propensity $f_1(s; \theta_1)$ and class posterior $f_2(s; \theta_2)$ via a dual-head network with coordination and discriminative regularization. High-confidence PU predictions further guide pseudo-centroid estimation for adaptive prototype calibration.

At Random (SCAR) assumption, yielding unbiased risk estimators such as uPU and nnPU (Du Plessis et al., 2015; Kiryo et al., 2017). Subsequent extensions enhance robustness through improved representation learning or reduced dependence on class-prior estimation (Chen et al., 2020a; Shi et al., 2018; Chen et al., 2020b). However, SCAR-based formulations neglect feature-dependent labeling bias and often fail to capture realistic annotation processes. To capture such feature-dependent labeling, instance-dependent PU learning explicitly models labeling probability as a function of input features. Early approaches employ Bayesian optimal relabeling (He et al., 2018) and EM-based latent-variable inference (Bekker & Davis, 2018) for probabilistic relabeling, while the Labeling Bias Estimation (LBE) framework (Gong et al., 2021) introduces a likelihood-based formulation that decouples the labeling mechanism from the class posterior without requiring prior class proportions. Building upon this paradigm, recent studies further incorporate contrastive or meta-learning strategies to handle more complex data regimes (Long et al., 2024; Kumar & Lambert, 2024).

Despite these advances, instance-dependent PU learning remains under-explored as a mechanism for domain-shift in GZSL, presenting a critical gap in mitigating the distribution bias between seen and unseen categories.

## 3. The Proposed PUFE Framework

The fundamental challenge in GZSL arises from the distribution shift between seen-class training data and the mixed-domain test data. The PUFE framework tackles this challenge through two synergistic modules: (i) a Semantic-Space PU classifier designed to disentangle high-confidence pseudo seen/unseen instances from the test set; and (ii) an adaptive semantic-calibration module that, guided by these filtered pseudo-instances, recalibrates category prototypes toward the semantic distribution underlying the test set.

### 3.1. Semantic-Space PU Classifier

Building upon a pretrained **embedding-based** GZSL baseline, we construct the semantic-space PU classification framework. Consider a semantic space $\mathcal{S} \subseteq \mathbb{R}^d$, where semantic representations are derived from the baseline. We define the training set $S_P$ and the test set $S_U$ as:

$$S_P = \{\boldsymbol{s}_1, \boldsymbol{s}_2, \ldots, \boldsymbol{s}_n\} \tag{1}$$

$$S_U = \{\boldsymbol{s}_{n+1}, \boldsymbol{s}_{n+2}, \ldots, \boldsymbol{s}_{n+m}\} \tag{2}$$

Here, $\boldsymbol{s}_i \in \mathcal{S}$ denotes the semantic embedding of the $i$-th sample. Here, $S_P$ comprises exclusively seen-class samples, whereas $S_U$ consists of an unlabeled mixture spanning both category sets.

The Semantic-Space PU classifier is designed to effectively disentangle seen and unseen categories within the unlabeled set $S_U$. We treat seen categories as positive $(+1)$ and unseen categories as negative $(-1)$. Specifically, each sample $\boldsymbol{s}_i \in S_P$ possesses a positive binary label $\hat{y}_i = 1$, whereas the latent binary labels of samples in $S_U$ belong to the set $\{+1, -1\}$.

We construct the classifier by modeling two conditional distributions: $p(\hat{a} \mid \hat{y}, \boldsymbol{s})$ and $p(\hat{y} \mid \boldsymbol{s})$. Here, $\hat{a} \in \{1, -1\}$ denotes the labeling status (where 1 represents labeled and $-1$ represents unlabeled), and $\hat{y}$ denotes binary label. Utilizing the sigmoid function $\sigma(z) = \frac{1}{1+e^{-z}}$, these distributions

are parameterized as:

$$p(\hat{a} = 1 \mid \hat{y} = 1, \boldsymbol{s}; \theta_1) = \sigma\big(f_1(\boldsymbol{s}; \theta_1)\big), \qquad (3)$$

$$p(\hat{y} = 1 \mid \boldsymbol{s}; \theta_2) = \sigma\big(f_2(\boldsymbol{s}; \theta_2)\big). \qquad (4)$$

where $\theta_1$ and $\theta_2$ are learnable parameters. Based on the positive-unlabeled (PU) setting assumption that negative samples are never labeled (i.e., $p(\hat{a} = 1 \mid \hat{y} = -1) = 0$), we derive the following complement probabilities:

$$p(\hat{a} = -1 \mid \hat{y} = 1, \boldsymbol{s}; \theta_1) = 1 - \sigma(f_1(\boldsymbol{s}; \theta_1)),$$
$$p(\hat{a} = -1 \mid \hat{y} = -1, \boldsymbol{s}; \theta_1) = 1, \qquad (5)$$
$$p(\hat{y} = -1 \mid \boldsymbol{s}; \theta_2) = 1 - \sigma(f_2(\boldsymbol{s}; \theta_2)).$$

**MLE-based Loss.** Given that samples in $S_P$ are observed as labeled ($\hat{a} = 1$) while those in $S_U$ remain unlabeled ($\hat{a} = -1$), the joint likelihood over $S_P \cup S_U$ is expressed as:

$$\mathcal{J} = \prod_{\boldsymbol{s} \in S_P} p(\hat{a} = 1, \hat{y} = 1 \mid \boldsymbol{s}; \theta_1, \theta_2)$$
$$\prod_{\boldsymbol{s} \in S_U} p(\hat{a} = -1 \mid \boldsymbol{s}; \theta_1, \theta_2). \qquad (6)$$

The components of the likelihood are factorized as:

$$p(\hat{a} = 1, \hat{y} = 1 \mid \boldsymbol{s}) = p(\hat{a} = 1 \mid \hat{y} = 1, \boldsymbol{s})p(\hat{y} = 1 \mid \boldsymbol{s}),$$
$$p(\hat{a} = -1 \mid \boldsymbol{s}) = \sum_{\hat{y} \in \{1, -1\}} p(\hat{a} = -1 \mid \hat{y}, \boldsymbol{s})p(\hat{y} \mid \boldsymbol{s}). \qquad (7)$$

Parameters $\theta_1$ and $\theta_2$ are estimated by maximizing Eq. (6). Substituting Eq. (3), (4), and (5) into the objective, and recognizing that $p(\hat{a} = -1 \mid \boldsymbol{s}) = 1 - p(\hat{a} = 1 \mid \boldsymbol{s})$, yields:

$$\max_{\theta_1, \theta_2} \prod_{\boldsymbol{s} \in S_P} \Big[\sigma(f_1(\boldsymbol{s}; \theta_1))\sigma(f_2(\boldsymbol{s}; \theta_2))\Big] \cdot$$
$$\prod_{\boldsymbol{s} \in S_U} \Big[1 - \sigma(f_1(\boldsymbol{s}; \theta_1))\sigma(f_2(\boldsymbol{s}; \theta_2))\Big]. \qquad (8)$$

Maximizing this likelihood is equivalent to minimizing the negative log-likelihood:

$$\min_{\theta_1, \theta_2} \sum_{\boldsymbol{s} \in S_P} -\log\Big[\sigma(f_1(\boldsymbol{s}; \theta_1))\sigma(f_2(\boldsymbol{s}; \theta_2))\Big]$$
$$+ \sum_{\boldsymbol{s} \in S_U} -\log\Big[1 - \sigma(f_1(\boldsymbol{s}; \theta_1))\sigma(f_2(\boldsymbol{s}; \theta_2))\Big]. \qquad (9)$$

To account for the size disparity between $S_P$ and $S_U$, we normalize each summation by its set size, recasting the objective into the following expectation-form PU risk:

$$\hat{R}_{pu} = \mathbb{E}_{\boldsymbol{s} \in S_P}\Big[l_p(f_1(\boldsymbol{s}; \theta_1), f_2(\boldsymbol{s}; \theta_2))\Big]$$
$$+ \mathbb{E}_{\boldsymbol{s} \in S_U}\Big[l_u(f_1(\boldsymbol{s}; \theta_1), f_2(\boldsymbol{s}; \theta_2))\Big]. \qquad (10)$$

where the component-wise loss functions are defined as:

$$l_p(f_1, f_2) = -\log\big(\sigma(f_1(\boldsymbol{s}))\sigma(f_2(\boldsymbol{s}))\big),$$
$$l_u(f_1, f_2) = -\log\big(1 - \sigma(f_1(\boldsymbol{s}))\sigma(f_2(\boldsymbol{s}))\big). \qquad (11)$$

**Coordination Regularization.** Minimizing Eq. (10) essentially treats the product term $\sigma(f_1(\boldsymbol{s}))\sigma(f_2(\boldsymbol{s}))$ as a single scalar, potentially failing to capture the intrinsic relationship between the propensity $f_1(\boldsymbol{s})$ and the prediction $f_2(\boldsymbol{s})$. Intuitively, $p(\hat{a} = 1 \mid \hat{y} = 1, \boldsymbol{s})$ and $p(\hat{y} = 1 \mid \boldsymbol{s})$ are expected to exhibit a positive correlation. In particular, a higher posterior probability $p(\hat{y} = 1 \mid \boldsymbol{s})$ indicates a higher confidence that $\boldsymbol{s}$ belongs to the positive class, which suggests that the labeling propensity $f_1(\boldsymbol{s})$ should also be high for such instances, motivating the consistency constraint between $f_1$ and $f_2$. To this end, we introduce a coordination regularization loss that imposes consistency between them:

$$\hat{R}_{\text{coord}} = \mathbb{E}_{\boldsymbol{s} \in S_P \cup S_U}\left[l_{\text{coord}}(f_1(\boldsymbol{s}; \theta_1), f_2(\boldsymbol{s}; \theta_2))\right]. \quad (12)$$

where

$$l_{\text{coord}}(f_1(\boldsymbol{s}; \theta_1), f_2(\boldsymbol{s}; \theta_2)) = (\sigma(f_1(\boldsymbol{s}; \theta_1)) - \sigma(f_2(\boldsymbol{s}; \theta_2)))^2 \qquad (13)$$

**Discriminative Geometry Regularization.** In GZSL, severe domain shift often causes unseen-class instances in $S_U$ to gravitate toward the high-density regions of seen categories, resulting in ambiguous decision boundaries. We therefore introduce discriminative geometry regularization to shape the posterior distribution $p(\hat{y} = 1 \mid \boldsymbol{s})$ by enforcing separability between the seen-class instances $S_P$ and the mixed unlabeled instances $S_U$.

Considering that $S_P$ is composed exclusively of seen-class instances while $S_U$ is a mixture of seen and unseen instances, the average confidence of $S_P$ belonging to the seen categories (positive class) should theoretically exceed that of $S_U$. To formalize this, we minimize the following separation term:

$$\hat{R}_{\text{sep}} = -\Big(\mathbb{E}_{\boldsymbol{s} \in S_P}[\sigma(f_2(\boldsymbol{s}; \theta_2))] - \mathbb{E}_{\boldsymbol{s} \in S_U}[\sigma(f_2(\boldsymbol{s}; \theta_2))]\Big). \qquad (14)$$

Equivalently, the distributional margin between the expected posteriors of $S_P$ and $S_U$ is maximized. Crucially, while $\hat{R}_{\text{sep}}$ exerts a global suppression pressure on $S_U$, the positive risk component of the PU objective acts as a strong anchor, forcing labeled samples in $S_P$ toward high posterior probabilities. Consequently, the latent seen instances within $S_U$, which are semantically clustered with $S_P$, are structurally resistant to this suppression due to the smoothness of the decision boundary (semantic continuity). In contrast, unseen instances, being distributionally distant from the dense core of $S_P$, lack such geometric support and are effectively suppressed toward zero. Thus, $\hat{R}_{\text{sep}}$ functions as a geometry-aware margin maximizer that selectively suppresses unseen categories.

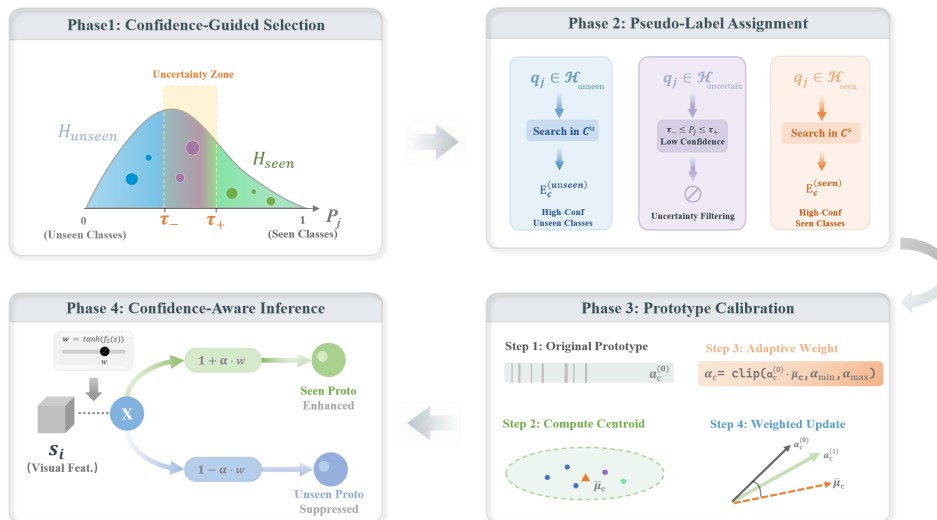

*Figure 3.* Schematic illustration of the adaptive multi-sample semantic calibration method.

Furthermore, to sharpen the decision boundaries, we impose a compactness constraint on $S_P$ to minimize the intra-class variance:

$$\hat{R}_{\mathrm{com}} = \mathrm{Var}_{\boldsymbol{s} \in S_P}[\sigma(f_2(\boldsymbol{s}; \theta_2))]. \tag{15}$$

The discriminative geometry regularization risk is then defined as:

$$\hat{R}_{\mathrm{geo}} = \hat{R}_{\mathrm{sep}} + \lambda_{\mathrm{com}} \hat{R}_{\mathrm{com}}, \tag{16}$$

where $\lambda_{\mathrm{com}} > 0$ is a fixed scaling constant set to 0.1 to balance the scales of $\hat{R}_{\mathrm{sep}}$ and $\hat{R}_{\mathrm{com}}$. Minimizing $\hat{R}_{\mathrm{geo}}$ encourages a well-structured posterior distribution by enhancing seen–unseen separability while preserving coherence within the seen-class distribution.

**Final Optimization Objective.** Combining the PU likelihood with the proposed regularization terms, the overall risk of PUFE is defined as

$$\min_{\theta_1, \theta_2} \hat{R}_{\mathrm{total}} = \hat{R}_{pu} + \lambda_{\mathrm{coord}} \hat{R}_{\mathrm{coord}} + \lambda_{\mathrm{geo}} \hat{R}_{\mathrm{geo}}, \tag{17}$$

where $\lambda_{\mathrm{coord}}, \lambda_{\mathrm{geo}} > 0$ are trade-off hyper-parameters.

### 3.2. Adaptive Semantic Calibration

This module refines prototypes by fusing confidence-guided pseudo-centroids into the original prototypes to better reflect the underlying test distribution, as illustrated in Figure 3.

**Confidence-Guided Pseudo-Instance Selection.** Using posterior statistics $\mu_p, \sigma_p$ over $S_U$, we handle distribution asymmetry via independent scalars $\kappa_{\pm} > 0$ to define thresholds $\tau_+ = \mu_p + \kappa_+ \sigma_p$ and $\tau_- = \mu_p - \kappa_- \sigma_p$. High-confidence subsets are selected by:

$$\begin{aligned} \mathcal{H}_{\mathrm{seen}} &= \{\boldsymbol{q} \mid p(\hat{y} = 1|\boldsymbol{q}) > \tau_+\}, \\ \mathcal{H}_{\mathrm{unseen}} &= \{\boldsymbol{q} \mid p(\hat{y} = 1|\boldsymbol{q}) < \tau_-\}. \end{aligned} \tag{18}$$

Let $\mathcal{C}^s$ and $\mathcal{C}^u$ denote the sets of seen and unseen categories, respectively. Each category $c$ is associated with a normalized semantic prototype $\boldsymbol{a}_c^{(0)}$ (where $\left\|\boldsymbol{a}_c^{(0)}\right\|_2 = 1$). We define $\mathrm{sim}(\cdot, \cdot)$ as the similarity metric (e.g., cosine similarity or dot product) adopted by the baseline framework. Accordingly, the pseudo-label $\hat{c}(\boldsymbol{q})$ for each $\boldsymbol{q}$ is determined by the most similar prototype within the search space: For $\boldsymbol{q} \in \mathcal{H}_{\mathrm{seen}}$,

$$\hat{c}(\boldsymbol{q}) = \arg\max_{c \in \mathcal{C}^s} \mathrm{sim}(\boldsymbol{q}, \boldsymbol{a}_c^{(0)}) \tag{19}$$

For $\boldsymbol{q} \in \mathcal{H}_{\mathrm{unseen}}$,

$$\hat{c}(\boldsymbol{q}) = \arg\max_{c \in \mathcal{C}^u} \mathrm{sim}(\boldsymbol{q}, \boldsymbol{a}_c^{(0)}) \tag{20}$$

Consequently, the resulting support set for each category $c$ is defined as:

$$E_c = \{\boldsymbol{q} \in \mathcal{H}_{\mathrm{seen}} \cup \mathcal{H}_{\mathrm{unseen}} \mid \hat{c}(\boldsymbol{q}) = c\} \tag{21}$$

**Prototype Calibration via Pseudo-Semantic Centroids.** To ensure statistical reliability, calibration is restricted to categories satisfying $|E_c| \geq N_{\mathrm{min}}$ (with $N_{\mathrm{min}} = 3$ to exclude unreliable estimates), updating the prototype $\boldsymbol{a}_c^{(0)}$ via:

$$\boldsymbol{a}_c^{(1)} = \frac{\alpha_c \boldsymbol{a}_c^{(0)} + (1 - \alpha_c)\bar{\boldsymbol{\mu}}_c}{\left\|\alpha_c \boldsymbol{a}_c^{(0)} + (1 - \alpha_c)\bar{\boldsymbol{\mu}}_c\right\|_2}, \tag{22}$$

where $\bar{\boldsymbol{\mu}}_c = \boldsymbol{\mu}_c / \|\boldsymbol{\mu}_c\|_2$ is the normalized pseudo-semantic centroid derived from the mean vector $\boldsymbol{\mu}_c = \frac{1}{|E_c|} \sum_{\boldsymbol{q} \in E_c} \boldsymbol{q}$. The fusion weight $\alpha_c$ is adaptively determined by the cosine similarity between the original prototype and the pseudo-centroid:

$$\alpha_c = \max\left(\alpha_{\mathrm{min}}, \min\left(\boldsymbol{a}_c^{(0)\,T}\bar{\boldsymbol{\mu}}_c, \alpha_{\mathrm{max}}\right)\right). \tag{23}$$

*Table 1.* GZSL performance with PUFE integration across seven backbone architectures.

| Method | CUB | | | | SUN | | | | AWA2 | | | |
|---|---|---|---|---|---|---|---|---|---|---|---|---|
| | U (%) | S (%) | H (%) | ΔH | U (%) | S (%) | H (%) | ΔH | U (%) | S (%) | H (%) | ΔH |
| TransZero (AAAI'22) | 69.3 | 68.3 | 68.8 | – | 52.6 | 33.4 | 40.8 | – | 61.3 | 82.3 | 70.2 | – |
| TransZero + PUFE | 73.7 | 76.8 | **75.2** | +6.4 | 46.6 | 39.1 | **42.5** | +1.7 | 71.0 | 93.2 | **80.6** | +10.4$^\dagger$ |
| MSDN (CVPR'22) | 68.7 | 67.5 | 68.1 | – | 52.2 | 34.2 | 41.3 | – | 62.0 | 74.5 | 67.7 | – |
| MSDN + PUFE | 78.4 | 74.3 | **76.3** | +8.2$^\dagger$ | 52.1 | 37.5 | **43.6** | +2.3 | 65.4 | 84.1 | **73.6** | +5.9 |
| IFMVZSL (ASC'24) | 62.8 | 64.5 | 63.6 | – | 46.4 | 33.9 | 39.2 | – | 79.6 | 76.8 | 78.2 | – |
| IFMVZSL + PUFE | 67.6 | 70.4 | **68.9** | +5.3 | 66.7 | 31.6 | **42.8** | +3.6 | 89.2 | 75.9 | **82.0** | +3.8 |
| PSVMA (CVPR'23) | 70.1 | 77.8 | 73.8 | – | 61.7 | 45.3 | 52.3 | – | 73.6 | 77.3 | 75.4 | – |
| PSVMA + PUFE | 83.8 | 84.5 | **84.1** | +10.3$^\dagger$ | 61.9 | 44.6 | **51.8** | −0.5 | 80.4 | 93.8 | **86.6** | +11.2$^\dagger$ |
| ZSLViT (CVPR'24) | 69.4 | 78.2 | 73.6 | – | 45.9 | 48.4 | 47.3 | – | 66.1 | 84.6 | 74.2 | – |
| ZSLViT + PUFE | 79.1 | 77.9 | **78.5** | +4.9 | 57.0 | 42.1 | **48.4** | +1.1 | 74.8 | 95.2 | **83.8** | +9.6$^\dagger$ |
| ZeroMamba (AAAI'25) | 72.1 | 76.4 | 74.2 | – | 56.5 | 41.4 | 47.7 | – | 67.9 | 87.6 | 76.5 | – |
| ZeroMamba + PUFE | 74.4 | 83.7 | **78.8** | +4.6 | 59.8 | 41.6 | **49.1** | +1.4 | 78.3 | 97.7 | **86.9** | +10.4$^\dagger$ |
| TPR (NeurIPS'24) | 26.9 | 41.2 | 32.5 | – | 45.4 | 50.5 | 47.8 | – | 76.8 | 87.1 | 81.6 | – |
| TPR + PUFE | 35.6 | 45.2 | **39.9** | +7.4 | 48.2 | 51.0 | **49.6** | +1.8 | 85.2 | 85.0 | **85.1** | +3.5 |

All results are rounded to one decimal. **Shaded rows**: original inductive baselines; **Unshaded rows**: same baselines integrated with PUFE.

Red: Best $H$; Blue: 2nd Best $H$; **Bold**: best in pair; $^\dagger$: $\Delta H > 8 points$.

The hyperparameters are determined via cross-validation as $\alpha_{\min} = 0.3$ and $\alpha_{\max} = 0.7$. The clipping mechanism balances stability and adaptivity: when the pseudo-centroid stays near the original prototype ($\alpha_c$ large), the reliable prior is preserved; when they diverge ($\alpha_c$ small), the lower bound limits overcorrection and prevents semantic drift.

Finally, we adopt a confidence-guided inference strategy to further refine predictions; the complete procedure and theoretical analysis are provided in Appendix C.

### 3.3. Optimization Strategy

In practice, we employ a decoupled two-stage strategy. First, the dual networks ($f_1, f_2$), parameterized as lightweight MLPs, are jointly optimized via stochastic gradient descent on $\hat{\mathcal{R}}_{\text{total}}$ Eq. (17). Subsequently, with learned parameters $\theta_1$ and $\theta_2$ fixed, prototypes are calibrated via the closed-form update (Eq. 22). More details and the pseudocode are provided in Appendix B.

## 4. Theoretical Analysis

We theoretically validate PUFE by establishing a uniform generalization risk bound for $\hat{\mathcal{R}}_{\text{total}}$ Eq. (17) and an effectiveness guarantee for the prototype calibration mechanism.

### 4.1. Generalization Bound for PU Learning.

We first present the generalization risk bound in the following theorem:

**Theorem 4.1** (Generalization Bound). *Let $\mathcal{F}$ be the hypothesis space of the labeling propensity network $f_1$ and the class posterior network $f_2$ (l-layer MLPs with bounded parameter norms, i.e., $\|\theta_1^{(i)}\|_F \leq M_{f_1}^{(i)}$ and $\|\theta_2^{(i)}\|_F \leq M_{f_2}^{(i)}$). Assume the activation functions are element-wise 1-Lipschitz continuous and positive-homogeneous, and that the semantic embedding of any instance is norm-bounded, i.e., $\|s\|_2 \leq B$. If the loss values are upper bounded by $A$, then for any $\delta > 0$, with probability at least $1 - \delta$, the gap between expected and empirical risk is bounded by:*

$$\left| R_{total}(f_1, f_2) - \hat{R}_{total}(f_1, f_2) \right|$$
$$\leq 3\sqrt{\frac{\log(2/\delta)}{2}} \left( \frac{\lambda_{coord}}{\sqrt{n+m}} + \frac{\lambda_{geo} + A}{\sqrt{m}} + \frac{3\lambda_{com}\lambda_{geo} + \lambda_{geo} + A}{\sqrt{n}} \right)$$
$$+ 2B(\sqrt{2l\log 2} + 1) \prod_{i=1}^{l} M_{f_1}^{(i)} \left( \frac{1}{\sqrt{n}} + \frac{1}{\sqrt{m}} + \frac{2\lambda_{coord}}{\sqrt{n+m}} \right)$$
$$+ B(\sqrt{2l\log 2} + 1) \prod_{i=1}^{l} M_{f_2}^{(i)} \left( \frac{108 + (86\lambda_{com} + 27)\lambda_{geo}}{54\sqrt{n}} + \frac{4 + \lambda_{geo}}{2\sqrt{m}} + \frac{4\lambda_{coord}}{\sqrt{n+m}} \right).$$
$$(24)$$

*where $M_{f_1}^{(i)}$ and $M_{f_2}^{(i)}$ bound the parameter norms of the i-th layer for $f_1$ and $f_2$, and $n, m$ denote the sizes of $S_P$ and $S_U$, respectively.*

Theorem 4.1 guarantees that the generalization risk of the learned $f_1$ and $f_2$ converges to the empirical risk at a rate of $\mathcal{O}(1/\sqrt{n} + 1/\sqrt{m} + 1/\sqrt{n+m})$ by increasing the sizes of $S_P$ and $S_U$ (see proof in Appendix A).

### 4.2. Theoretical Guarantee for Calibration.

We further verify theoretically that the proposed adaptive calibration mechanism strictly improves the quality of semantic prototypes under mild conditions.

**Theorem 4.2** (Calibration Guarantee). *Let $\boldsymbol{\mu}_c^*$ be the ground-truth class center. Assuming the pseudo-centroid $\bar{\boldsymbol{\mu}}_c$ provides a superior estimation to the original prototype (see Lemma C.1), the calibrated prototype strictly improves semantic alignment, i.e., $\mathrm{sim}(\boldsymbol{a}_c^{(1)}, \boldsymbol{\mu}_c^*) > \mathrm{sim}(\boldsymbol{a}_c^{(0)}, \boldsymbol{\mu}_c^*)$*

*Table 2.* Plug-in method comparison under identical backbone and evaluation conditions.

| Base | Method | Set. | CUB | | | SUN | | | AWA2 | | |
|------|--------|------|-----|-----|-----|-----|-----|-----|------|-----|-----|
| | | | U (%) | S (%) | H (%) | U (%) | S (%) | H (%) | U (%) | S (%) | H (%) |
| TransZero | Original (Chen et al., 2022b) | Ind. | 69.3 | 68.3 | 68.8 | 52.6 | 33.4 | 40.8 | 61.3 | 82.3 | 70.2 |
| | + STHS (Bo et al., 2021) | Tran. | 82.6 | 63.0 | 71.5 | 52.7 | 37.3 | **43.7** | 66.3 | 80.8 | 72.8 |
| | + CCC (Yue et al., 2024) | Tran. | 69.7 | 68.6 | 69.1 | 63.2 | 33.0 | 40.4 | 59.8 | 82.5 | 69.4 |
| | + PC (Bhat et al., 2025) | Cal. | 70.1 | 71.9 | 71.0 | 52.3 | 35.4 | 42.2 | 63.1 | 81.9 | 71.3 |
| | + PUFE (Ours) | Tran. | 73.7 | 76.8 | **75.2** | 46.6 | 39.1 | 42.5 | 71.0 | 93.2 | **80.6** |
| PSVMA | Original (Liu et al., 2023) | Ind. | 70.1 | 77.8 | 73.8 | 61.7 | 45.3 | 52.3 | 73.6 | 77.3 | 75.4 |
| | + STHS (Bo et al., 2021) | Tran. | 83.1 | 82.7 | 82.9 | 58.7 | 50.4 | 54.2 | 69.1 | 83.5 | 75.6 |
| | + CCC (Yue et al., 2024) | Tran. | 73.2 | 82.3 | 77.4 | 60.8 | 46.1 | 52.4 | 80.0 | 90.4 | 84.9 |
| | + PC (Bhat et al., 2025) | Cal. | 78.6 | 75.7 | 77.1 | 65.9 | 50.0 | **56.8** | 72.7 | 87.6 | 79.5 |
| | + PUFE (Ours) | Tran. | 83.8 | 84.5 | **84.1** | 61.9 | 44.6 | 51.8 | 80.4 | 93.8 | **86.6** |

All results are rounded to one decimal. **Ind.**: Inductive; **Tran.**: Transductive (TTA paradigm, mixed unlabeled test set, no unseen-class pool);

**Cal.**: Posterior Calibration (class-prior-based, no test-set access); **Bold**: Best $H$ per group.

*for any* $\alpha_c < 1$, *where* sim *denotes the baseline-dependent metric.*

Theorem 4.2 confirms that PUFE harnesses latent geometry to mitigate bias by pulling prototypes closer to their true cluster centers (proof in Appendix C.2). The guarantee is conditional on $\varepsilon_e < \varepsilon_s$, i.e., the PU-derived pseudo-centroid must be a better estimator of the true class center than the original prototype.

## 5. Experiments

### 5.1. Implementation Details

We evaluate PUFE on three benchmarks (CUB, SUN, AWA2) across seven representative GZSL baselines. Detailed descriptions of datasets and baseline architectures are provided in Appendix D. All experiments are implemented in PyTorch on a single NVIDIA RTX 4090 GPU. Semantic embeddings are normalized prior to PU learning. For each dataset, hyperparameters ($\lambda_{\text{coord}}$, $\lambda_{\text{geo}}$, $\kappa_+$, $\kappa_-$) are tuned on a validation split following the standard protocol (Xian et al., 2018a), where held-out seen classes are treated as proxy unseen classes to maintain strict zero-shot compliance. Detailed experimental data and statistical analyses are provided in Appendix E.

### 5.2. Performance Comparison on GZSL Benchmarks

**Efficacy of PUFE Integration.** As shown in Table 1, PUFE delivers consistent gains across all seven backbone architectures, spanning attention-based Transformers, state-space models, and VLM-based frameworks. Six backbone–dataset pairs surpass an 8-point rise in H — most notably PSVMA (+11.2 on AWA2 and +10.3 on CUB), alongside both TransZero and ZeroMamba (+10.4 on AWA2). PSVMA + PUFE sets a new state-of-the-art on CUB (84.1%), while ZeroMamba + PUFE achieves the highest reported $H$ on AWA2 (86.9%). The sole negative case is PSVMA on

SUN ($\Delta H = -0.5\%$). Since PSVMA already performs prototype-level alignment within its backbone to counteract seen-class bias, the margin left for post-hoc calibration is narrow, and SUN's sparsely annotated, highly overlapping attribute space leaves the condition $\varepsilon_e < \varepsilon_s$ of Theorem 4.2 only weakly satisfied.

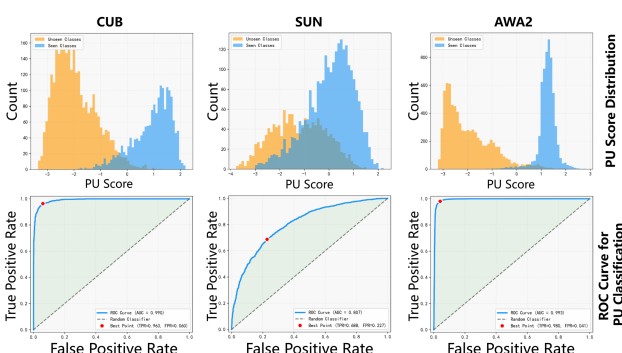

*Figure 4.* Quantitative Analysis of PU Classifier (ZSLViT+PUFE). **Top:** Histograms of raw PU scores (pre-sigmoid logits) exhibiting a distinct bimodal structure, where seen (blue) and unseen (orange) classes are clearly separated on CUB and AWA2. **Bottom:** ROC curves quantifying discriminative capability, with near-ideal AUCs confirming the reliability of our confidence-aware calibration.

**Comparison with Plug-in Methods.** Table 2 benchmarks PUFE against STHS (Bo et al., 2021), CCC (Yue et al., 2024), and PC (Bhat et al., 2025) under identical backbone conditions. PUFE achieves the best $H$ on CUB and AWA2 across both base models—e.g., 84.1% and 86.6% with PSVMA, outperforming all alternatives by a clear margin. On SUN, the inherently limited calibration headroom leaves methods built on test-data clustering (STHS) or class-prior correction (PC) better placed to achieve higher $H$ in this setting. Overall, PUFE attains the best $H$ in four of six comparison groups, establishing that explicit PU-based transductive test-time distribution modeling provides more principled and robust calibration than methods relying on cluster-structure assumptions or pre-estimated class priors.

## 5.3. Discriminative Capability of PU Classifier

Table 3 reports the discriminative performance of the PU classifier across five baselines, evaluated by Precision (Prec$^\pm$) and $F_1$-score ($F_1^\pm$) for both seen (+) and unseen (−) classes. The module attains near-perfect discrimination on both the coarse-grained AWA2 and fine-grained CUB datasets, with TransZero reaching 98.2% accuracy on AWA2 and PSVMA achieving 96.9% on CUB (with an $F_1^-$ of 0.972). Even on the challenging SUN dataset, where semantic overlap is prevalent, the framework maintains robust performance (e.g., 70.8% accuracy with ZS-LViT). This quantitative success is visually corroborated by Figure 4. The histograms (Top) display the distribution of raw PU scores (logits), revealing a bimodal structure on CUB and AWA2 where seen (blue) and unseen (orange) classes are well-separated. This distinct separation translates to near-ideal ROC curves (Bottom) with AUC $\approx 0.99$. By leveraging these reliable confidence scores, PUFE filters mid-range ambiguity, ensuring that prototype calibration is driven exclusively by high-quality instances.

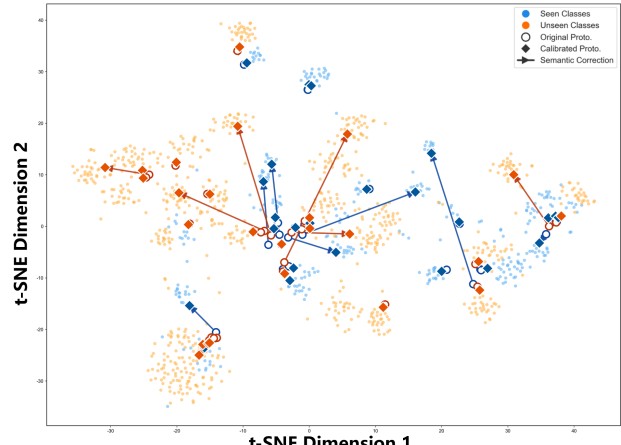

*Figure 5.* **Visualization of Prototype Alignment on CUB.** Vectors depict the evolution from original prototypes (circles) to calibrated ones (diamonds) for a randomly selected subset of 50 classes (25 seen and 25 unseen) to ensure visual clarity. The trajectories progressively correct each prototype's initial offset from its latent semantic cluster, illustrating the effect of adaptive calibration.

*Table 3.* PU classification performance based on baseline models

| Baseline | Dataset | Prec$^+$ | Prec$^-$ | $F_1^+$ | $F_1^-$ | Acc |
|---|---|---|---|---|---|---|
| **TransZero** | **CUB** | 0.780 | 0.922 | 0.827 | 0.886 | 0.863 |
| | **SUN** | 0.828 | 0.553 | 0.734 | 0.639 | 0.694 |
| | **AWA2** | 0.969 | 0.991 | 0.977 | 0.985 | **0.982** |
| **MSDN** | **CUB** | 0.866 | 0.937 | 0.881 | 0.927 | 0.909 |
| | **SUN** | 0.788 | 0.553 | 0.743 | 0.602 | 0.687 |
| | **AWA2** | 0.944 | 0.964 | 0.947 | 0.961 | 0.955 |
| **IFMVZSL** | **CUB** | 0.840 | 0.934 | 0.867 | 0.914 | 0.896 |
| | **SUN** | 0.802 | 0.571 | 0.756 | 0.623 | 0.703 |
| | **AWA2** | 0.941 | 0.976 | 0.954 | 0.966 | 0.961 |
| **PSVMA** | **CUB** | 0.947 | 0.985 | 0.963 | 0.972 | **0.969** |
| | **SUN** | 0.821 | 0.563 | 0.745 | 0.637 | 0.701 |
| | **AWA2** | 0.961 | 0.979 | 0.967 | 0.975 | 0.972 |
| **ZSLViT** | **CUB** | 0.829 | 0.953 | 0.875 | 0.919 | 0.902 |
| | **SUN** | 0.853 | 0.566 | 0.743 | 0.661 | **0.708** |
| | **AWA2** | 0.919 | 0.976 | 0.943 | 0.956 | 0.950 |

**Bold**: Best Acc. Prec$^+$/Prec$^-$ denote Positive/Negative Predictive Value.
$F_1^+ = \frac{2 \cdot \text{Prec}^+ \cdot \text{Recall}}{\text{Prec}^+ + \text{Recall}}$; $F_1^- = \frac{2 \cdot \text{Prec}^- \cdot \text{Specificity}}{\text{Prec}^- + \text{Specificity}}$.

## 5.4. Qualitative Visualization of Prototype Alignment

Figure 5 illustrates the calibration mechanism underlying the performance gains. A clear "centripetal motion" emerges: prototypes shift from sparse margins into high-density cluster cores. This trajectory, driven by PU-estimated pseudo-centroids, illustrates how PUFE exploits unlabeled structure to counteract domain shift and sharpen prototype boundaries.

## 5.5. Ablation Study

Table 4 dissects the contribution of each component within the PUFE framework using PSVMA as the baseline. Here, Base (LBE) refers to applying the MLE objective (Eq. 10)

alone. Integrating either semantic coordination (+C) or discriminative geometry regularization (+G) individually yields steady improvements. For instance, on AWA2, +C and +G improve the H-score from 84.0% to 85.0% and 84.9%, respectively, validating their individual contributions to posterior estimation quality. Combining both regularizers yields the best $H$ across all benchmarks — 86.6 on AWA2, 84.1 on CUB, and 51.8 on SUN — exceeding either component alone. The gain stems primarily from PU classification accuracy, which improves most on SUN while remaining near-saturation on AWA2 and CUB. These results confirm that jointly enforcing local consistency and geometric separability ensures reliable propensity estimation for superior GZSL generalization.

*Table 4.* Ablation Study: Progressive Component Analysis (in %).

| Method | AWA2 | | | | CUB | | | | SUN | | | |
|---|---|---|---|---|---|---|---|---|---|---|---|---|
| | Acc | U | S | H | Acc | U | S | H | Acc | U | S | H |
| **Base (LBE)** | 96.9 | 75.1 | 95.3 | 84.0 | 94.6 | 84.9 | 79.9 | 82.3 | 58.7 | 56.0 | 43.3 | 48.8 |
| **Base + C** | 96.9 | 76.7 | 95.2 | 85.0 | 95.4 | 82.0 | 84.8 | 83.4 | 67.2 | 55.8 | 42.8 | 48.5 |
| **Base + G** | 97.1 | 76.7 | 95.1 | 84.9 | 95.0 | 80.9 | 84.9 | 82.8 | 69.9 | 59.5 | 44.5 | 50.9 |
| **PUFE (Full)** | **97.2** | 80.4 | 93.8 | **86.6** | **96.9** | 83.8 | 84.5 | **84.1** | **70.1** | 61.9 | 44.6 | **51.8** |

All results are rounded to one decimal.

## 5.6. Parameter Sensitivity Analysis

Taking AWA2 as an example, we analyze the hyperparameter sensitivity of PUFE using TransZero.

**Coordination and Discriminative Weights ($\lambda_{\text{coord}}, \lambda_{\text{geo}}$).** Figure 6 illustrates the impact of regularization weights on GZSL performance. The $H$-score peaks at $\lambda_{\text{coord}} = 0.3$ and $\lambda_{\text{geo}} = 0.1$: as $\lambda_{\text{coord}}$ increases from 0.1 to 0.3, $U$ improves while $S$ stays stable, yielding the optimal $H$; excessively

large values disrupt this balance. Importantly, moderate variations in these weights lead to only marginal changes in $H$, indicating that PUFE is robust to hyperparameter selection and requires only coarse tuning.

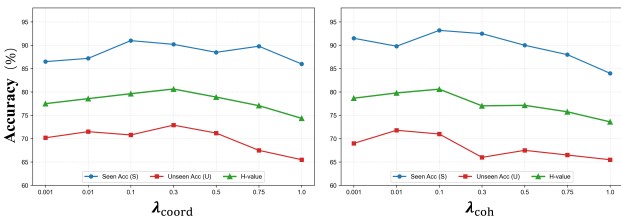

*Figure 6.* Parameter sensitivity analysis on AWA2. (a) Impact of coordination weight $\lambda_{\text{coord}}$. (b) Impact of discriminative weight $\lambda_{\text{geo}}$. The curves demonstrate that moderate regularization effectively balances seen and unseen recognition, yielding optimal GZSL performance.

**Adaptive Threshold Scaling Factors** $(\kappa_+, \kappa_-)$**.** Figure 7 visualizes the joint sensitivity of $\kappa_+$ and $\kappa_-$. The heatmap reveals an optimal operating point at $(\kappa_+, \kappa_-) = (1.0, 2.0)$, where the model achieves an $H$ of 80.6%. $H$ is more sensitive to $\kappa_-$ than $\kappa_+$, suggesting that high-confidence unseen samples play a more critical role in precise prototype calibration. The scaling factors for CUB and SUN were determined as $(0.5, 0.5)$ and $(1.5, 2.0)$, respectively. As a practical guideline, we recommend initializing with $(\lambda_{\text{coord}}, \lambda_{\text{geo}}) = (0.3, 0.1)$ and $(\kappa_+, \kappa_-) = (1.0, 2.0)$ as a reasonable starting point, with dataset-specific fine-tuning advised thereafter.

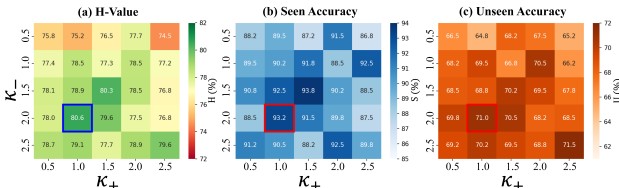

*Figure 7.* Joint impact heatmap of scaling factors $(\kappa_+, \kappa_-)$ on the validation performance of TransZero+PUFE on AWA2. Both axes denote the absolute values of $\kappa_+$ and $\kappa_-$. The highlighted cell marks the selected optimum $(\kappa_+, \kappa_-) = (1.0, 2.0)$ across all three panels.

## 6. Conclusion

In this work, we have presented PUFE, a test-time transductive framework that mitigates seen-class bias by recasting GZSL inference as a Positive-Unlabeled (PU) learning problem over the naturally mixed unlabeled test set. A dual-head network jointly estimates seen-class posteriors and labeling propensities via maximum likelihood estimation, coupled with coordination and discriminative geometry regularization. These posteriors then guide confidence-based pseudo-instance selection for adaptive prototype calibration against the test-time distribution shift. PUFE requires no backbone

retraining, integrating seamlessly as a lightweight plug-and-play module into existing embedding-based GZSL models, underpinned by theoretical guarantees on generalization and prototype alignment.

Empirical evaluations on CUB, SUN, and AWA2 validate the efficacy of the proposed PU-based inference reformulation. PUFE delivers notable gains in harmonic mean across seven diverse baselines, underscoring that mining posterior estimates from latent test structures is the key to effective prototype correction. Qualitative visualization and theoretical analysis indicate that PUFE reliably separates seen from unseen categories by pulling prototypes toward their true semantic centroids. Future work will explore streaming inference extensions under evolving test distributions (Liang et al., 2025; Wang et al., 2025), while further integrating large-scale self-supervised visual encoders such as DINOv2 (Oquab et al., 2023) to broaden the applicability of PUFE.

## Acknowledgements

The authors are grateful to the Area Chairs and the anonymous reviewers for their constructive comments.

## Impact Statement

This paper presents work whose goal is to advance the field of Machine Learning. There are many potential societal consequences of our work, none which we feel must be specifically highlighted here.

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

## A. Generalization Analysis

In this subsection, we provide the detailed proof for the generalization bound of the PUFE framework (Theorem 4.1). This analysis guarantees that the empirical risk minimization over the dual networks leads to a bounded expected risk.

According to the empirical risks (11) and (13), corresponding expected risks are defined as follows:

$$R_{pu}(f_1, f_2) = \mathbb{E}_{\boldsymbol{s} \sim \mathcal{D}_P} \left[ l_p \left( \sigma(f_1(\boldsymbol{s})) \sigma(f_2(\boldsymbol{s})) \right) \right] + \mathbb{E}_{\boldsymbol{s} \sim \mathcal{D}_U} \left[ l_u \left( \sigma(f_1(\boldsymbol{s})) \sigma(f_2(\boldsymbol{s})) \right) \right] \tag{25}$$

$$R_{\text{coord}}(f_1, f_2) = \mathbb{E}_{\boldsymbol{s} \sim \mathcal{D}_P \cup \mathcal{D}_U} \left[ l_{\text{coord}} \left( f_1(\boldsymbol{s}), f_2(\boldsymbol{s}) \right) \right] \tag{26}$$

$$R_{\text{geo}}(f_2) = -\mathbb{E}_{\boldsymbol{s} \sim \mathcal{D}_P} \left[ \sigma(f_2(\boldsymbol{s})) \right] + \mathbb{E}_{\boldsymbol{s} \sim \mathcal{D}_U} \left[ \sigma(f_2(\boldsymbol{s})) \right]$$
$$+ \lambda_{\text{com}} \left\{ \mathbb{E}_{\boldsymbol{s} \sim \mathcal{D}_P} \left[ \sigma(f_2(\boldsymbol{s}))^2 \right] - \left( \mathbb{E}_{\boldsymbol{s} \sim \mathcal{D}_P} \left[ \sigma(f_2(\boldsymbol{s})) \right] \right)^2 \right\} \tag{27}$$

$$R_{\text{total}}(f_1, f_2) = R_{pu}(f_1, f_2) + \lambda_{\text{coord}} R_{\text{coord}}(f_1, f_2) + \lambda_{\text{geo}} R_{\text{geo}}(f_2) \tag{28}$$

where $\mathcal{D}_P$ and $\mathcal{D}_U$ denote the underlying distributions of $S_P$ and $S_U$ respectively. For the generalization error bound of PUFE, we have the following theorem.

**Theorem 4.1 (Restated).** *Let the labeling propensity network $f_1(\boldsymbol{s}; \boldsymbol{\theta}_1)$ and the class posterior network $f_2(\boldsymbol{s}; \boldsymbol{\theta}_2)$ be l-layer MLP networks. Assume the Frobenius norms of the parameters in each layer $1 \leq i \leq l$ are bounded as $\left\| \boldsymbol{\theta}_1^{(i)} \right\|_F \leq M_{f_1}^{(i)}$ and $\left\| \boldsymbol{\theta}_2^{(i)} \right\|_F \leq M_{f_2}^{(i)}$, respectively. Assume the semantic projection of any instance $\boldsymbol{s}$ satisfies $\|\boldsymbol{s}\|_2 \leq B$. Furthermore, assume the activation functions are element-wise 1-Lipschitz continuous and positive-homogeneous (e.g., ReLU). If the loss values of $l_p$ and $l_u$ are upper bounded by $A$, then for any $0 < \delta < 1$, with probability at least $1 - \delta$, the following generalization bound holds:*

$$\left| R_{\text{total}}(f_1, f_2) - \hat{R}_{\text{total}}(f_1, f_2) \right| \leq 3\sqrt{\frac{\log(2/\delta)}{2}} \left( \frac{\lambda_{\text{coord}}}{\sqrt{n+m}} + \frac{\lambda_{\text{geo}} + A}{\sqrt{m}} + \frac{3\lambda_{\text{com}}\lambda_{\text{geo}} + \lambda_{\text{geo}} + A}{\sqrt{n}} \right)$$
$$+ 2B(\sqrt{2l\log 2} + 1) \prod_{i=1}^{l} M_{f_1}^{(i)} \left( \frac{1}{\sqrt{n}} + \frac{1}{\sqrt{m}} + \frac{2\lambda_{\text{coord}}}{\sqrt{n+m}} \right)$$
$$+ B(\sqrt{2l\log 2} + 1) \prod_{i=1}^{l} M_{f_2}^{(i)} \left( \frac{108 + (86\lambda_{\text{com}} + 27)\lambda_{\text{geo}}}{54\sqrt{n}} + \frac{4 + \lambda_{\text{geo}}}{2\sqrt{m}} + \frac{4\lambda_{\text{coord}}}{\sqrt{n+m}} \right) \tag{29}$$

*where $n, m$ denote the sizes of the seen set $S_P$ and unlabeled set $S_U$, respectively, and $M_{f_1}^{(i)}, M_{f_2}^{(i)}$ are the Frobenius norm upper bounds for the weight parameters of the $i$-th layer in networks $f_1$ and $f_2$.*

**Proof.** Based on the empirical Rademacher complexity (Bartlett & Mendelson, 2002), we define:

$$\hat{\mathcal{R}}_p \left( l_p^{\circ}(\mathcal{F} \times \mathcal{F}) \right) = \mathbb{E}_{\boldsymbol{r}} \left[ \sup_{f_1, f_2 \in \mathcal{F}} \frac{1}{n} \sum_{i=1}^{n} r_i l_p \left( f_1(\boldsymbol{s}_i), f_2(\boldsymbol{s}_i) \right) \right], \tag{30}$$

$$\hat{\mathcal{R}}_u \left( l_u^{\circ}(\mathcal{F} \times \mathcal{F}) \right) = \mathbb{E}_{\boldsymbol{r}} \left[ \sup_{f_1, f_2 \in \mathcal{F}} \frac{1}{m} \sum_{i=n+1}^{n+m} r_i l_u \left( f_1(\boldsymbol{s}_i), f_2(\boldsymbol{s}_i) \right) \right], \tag{31}$$

$$\hat{\mathcal{R}} \left( l_{\text{coord}}^{\circ}(\mathcal{F} \times \mathcal{F}) \right) = \mathbb{E}_{\boldsymbol{r}} \left[ \sup_{f_1, f_2 \in \mathcal{F}} \frac{1}{n+m} \sum_{i=1}^{n+m} r_i l_{\text{coord}} \left( f_1(\boldsymbol{s}_i), f_2(\boldsymbol{s}_i) \right) \right], \tag{32}$$

$$\hat{\mathcal{R}}_p \left( \sigma^{\circ} \mathcal{F} \right) = \mathbb{E}_{\boldsymbol{r}} \left[ \sup_{f_2 \in \mathcal{F}} \frac{1}{n} \sum_{i=1}^{n} r_i \sigma \left( f_2(\boldsymbol{s}_i) \right) \right], \tag{33}$$

$$\hat{\mathcal{R}}_u \left( \sigma^{\circ} \mathcal{F} \right) = \mathbb{E}_{\boldsymbol{r}} \left[ \sup_{f_2 \in \mathcal{F}} \frac{1}{m} \sum_{i=n+1}^{n+m} r_i \sigma \left( f_2(\boldsymbol{s}_i) \right) \right], \tag{34}$$

$$\hat{\mathcal{R}}_p \left( \sigma^{2\circ} \mathcal{F} \right) = \mathbb{E}_{\boldsymbol{r}} \left[ \sup_{f_2 \in \mathcal{F}} \frac{1}{n} \sum_{i=1}^{n} r_i \sigma \left( f_2 \left( \boldsymbol{s}_i \right) \right)^2 \right], \tag{35}$$

where each $r_i$ is an independent Rademacher variable uniformly drawn from $\{1, -1\}$.

Further, we have:

$$\left| \frac{\partial l_p \left( f_1, f_2 \right)}{\partial f_1} \right| = \left| -\frac{1}{\sigma \left( f_1 \right) \sigma \left( f_2 \right)} \sigma \left( f_2 \right) \sigma \left( f_1 \right) \left( 1 - \sigma \left( f_1 \right) \right) \right| = \left| 1 - \sigma \left( f_1 \right) \right| \leq 1 \tag{36}$$

$$\left| \frac{\partial l_p \left( f_1, f_2 \right)}{\partial f_2} \right| = \left| -\frac{1}{\sigma \left( f_1 \right) \sigma \left( f_2 \right)} \sigma \left( f_1 \right) \sigma \left( f_2 \right) \left( 1 - \sigma \left( f_2 \right) \right) \right| = \left| 1 - \sigma \left( f_2 \right) \right| \leq 1 \tag{37}$$

$$\left| \frac{\partial l_u \left( f_1, f_2 \right)}{\partial f_1} \right| = \left| \frac{1}{1 - \sigma \left( f_1 \right) \sigma \left( f_2 \right)} \sigma \left( f_2 \right) \sigma \left( f_1 \right) \left( 1 - \sigma \left( f_1 \right) \right) \right|$$
$$\leq \left| \frac{1}{1 - \sigma \left( f_1 \right)} \sigma \left( f_2 \right) \sigma \left( f_1 \right) \left( 1 - \sigma \left( f_1 \right) \right) \right| = \left| \sigma \left( f_2 \right) \sigma \left( f_1 \right) \right| \leq 1 \tag{38}$$

$$\left| \frac{\partial l_u \left( f_1, f_2 \right)}{\partial f_2} \right| = \left| \frac{1}{1 - \sigma \left( f_1 \right) \sigma \left( f_2 \right)} \sigma \left( f_1 \right) \sigma \left( f_2 \right) \left( 1 - \sigma \left( f_2 \right) \right) \right|$$
$$\leq \left| \frac{1}{1 - \sigma \left( f_2 \right)} \sigma \left( f_1 \right) \sigma \left( f_2 \right) \left( 1 - \sigma \left( f_2 \right) \right) \right| = \left| \sigma \left( f_1 \right) \sigma \left( f_2 \right) \right| \leq 1 \tag{39}$$

$$\left| \frac{\partial l_{\text{coord}} \left( f_1, f_2 \right)}{\partial f_1} \right| = \left| 2 \left( \sigma \left( f_1 \right) - \sigma \left( f_2 \right) \right) \right| \leq 2 \tag{40}$$

$$\left| \frac{\partial l_{\text{coord}} \left( f_1, f_2 \right)}{\partial f_2} \right| = \left| 2 \left( \sigma \left( f_2 \right) - \sigma \left( f_1 \right) \right) \right| \leq 2 \tag{41}$$

$$\left| \frac{\partial \sigma \left( f_2 \right)}{\partial f_2} \right| = \left| \sigma \left( f_2 \right) \left( 1 - \sigma \left( f_2 \right) \right) \right| \leq \frac{1}{4} \tag{42}$$

$$\left| \frac{\partial \left( \sigma \left( f_2 \right) \right)^2}{\partial f_2} \right| = \left| 2 \left( \sigma \left( f_2 \right) \right)^2 \left( 1 - \sigma \left( f_2 \right) \right) \right| \leq \frac{8}{27} \tag{43}$$

According to Eq. (36)-(37), $l_p \left( f_1, f_2 \right)$ is $L$-Lipschitz ($L = 1$) continuous with respect to $f_1$ or $f_2$. Thus, we have (Ledoux & Talagrand, 1991):

$$\hat{\mathcal{R}}_p \left( l_p^\circ (\mathcal{F} \times \mathcal{F}) \right) \leq L \mathbb{E}_{\boldsymbol{r}} \left[ \sup_{f_1 \in \mathcal{F}} \frac{1}{n} \sum_{i=1}^{n} r_i f_1 \left( \boldsymbol{s}_i \right) \right] + L \mathbb{E}_{\boldsymbol{r}} \left[ \sup_{f_2 \in \mathcal{F}} \frac{1}{n} \sum_{i=1}^{n} r_i f_2 \left( \boldsymbol{s}_i \right) \right]$$
$$= \mathbb{E}_{\boldsymbol{r}} \left[ \sup_{f_1 \in \mathcal{F}} \frac{1}{n} \sum_{i=1}^{n} r_i f_1 \left( \boldsymbol{s}_i \right) \right] + \mathbb{E}_{\boldsymbol{r}} \left[ \sup_{f_2 \in \mathcal{F}} \frac{1}{n} \sum_{i=1}^{n} r_i f_2 \left( \boldsymbol{s}_i \right) \right] \tag{44}$$

Similarly, we can obtain:

$$\hat{\mathcal{R}}_u \left( l_u^\circ (\mathcal{F} \times \mathcal{F}) \right) \leq \mathbb{E}_{\boldsymbol{r}} \left[ \sup_{f_1 \in \mathcal{F}} \frac{1}{m} \sum_{i=n+1}^{n+m} r_i f_1 \left( \boldsymbol{s}_i \right) \right] + \mathbb{E}_{\boldsymbol{r}} \left[ \sup_{f_2 \in \mathcal{F}} \frac{1}{m} \sum_{i=n+1}^{n+m} r_i f_2 \left( \boldsymbol{s}_i \right) \right] \tag{45}$$

$$\hat{\mathcal{R}} \left( l_{\text{coord}}^\circ (\mathcal{F} \times \mathcal{F}) \right) \leq 2 \mathbb{E}_{\boldsymbol{r}} \left[ \sup_{f_1 \in \mathcal{F}} \frac{1}{n+m} \sum_{i=1}^{n+m} r_i f_1 \left( \boldsymbol{s}_i \right) \right] + 2 \mathbb{E}_{\boldsymbol{r}} \left[ \sup_{f_2 \in \mathcal{F}} \frac{1}{n+m} \sum_{i=1}^{n+m} r_i f_2 \left( \boldsymbol{s}_i \right) \right] \tag{46}$$

$$\hat{\mathcal{R}}_p \left( \sigma^\circ \mathcal{F} \right) \leq \frac{1}{4} \mathbb{E}_{\boldsymbol{r}} \left[ \sup_{f_2 \in \mathcal{F}} \frac{1}{n} \sum_{i=1}^{n} r_i f_2 \left( \boldsymbol{s}_i \right) \right] \tag{47}$$

$$\hat{\mathcal{R}}_u \left( \sigma^\circ \mathcal{F} \right) \leq \frac{1}{4} \mathbb{E}_{\boldsymbol{r}} \left[ \sup_{f_2 \in \mathcal{F}} \frac{1}{m} \sum_{i=n+1}^{n+m} r_i f_2 \left( \boldsymbol{s}_i \right) \right] \tag{48}$$

$$\hat{\mathcal{R}}_p \left( \sigma^{2\circ} \mathcal{F} \right) \leq \frac{8}{27} \mathbb{E}_{\boldsymbol{r}} \left[ \sup_{f_2 \in \mathcal{F}} \frac{1}{n} \sum_{i=1}^{n} r_i f_2 \left( \boldsymbol{s}_i \right) \right] \tag{49}$$

Invoking the standard generalization bound (Bartlett & Mendelson, 2002), for any $0 < \delta < 1$, the following inequalities hold with probability at least $1 - \delta$. First, for the PU risk component:

$$
\sup_{f_1, f_2 \in \mathcal{F}} \left| R_{pu}(f_1, f_2) - \hat{R}_{pu}(f_1, f_2) \right|
$$

$$
\leq \sup_{f_1, f_2 \in \mathcal{F}} \left| \mathbb{E}_{\boldsymbol{s} \sim \mathcal{D}_P} \left[ l_p(f_1(\boldsymbol{s}), f_2(\boldsymbol{s})) \right] - \frac{1}{n} \sum_{\boldsymbol{s} \in S_P} \left[ l_p(f_1(\boldsymbol{s}), f_2(\boldsymbol{s})) \right] \right|
$$

$$
+ \sup_{f_1, f_2 \in \mathcal{F}} \left| \mathbb{E}_{\boldsymbol{s} \sim \mathcal{D}_U} \left[ l_u(f_1(\boldsymbol{s}), f_2(\boldsymbol{s})) \right] - \frac{1}{m} \sum_{\boldsymbol{s} \in S_U} \left[ l_u(f_1(\boldsymbol{s}), f_2(\boldsymbol{s})) \right] \right|
$$

$$
\leq 2\hat{\mathcal{R}}_p \left( l_p^\circ(\mathcal{F} \times \mathcal{F}) \right) + 2\hat{\mathcal{R}}_u \left( l_u^\circ(\mathcal{F} \times \mathcal{F}) \right) + 3A\sqrt{\frac{\log \frac{2}{\delta}}{2n}} + 3A\sqrt{\frac{\log \frac{2}{\delta}}{2m}}
$$

$$(50)$$

Similarly, for the coordination risk:

$$
\sup_{f_1, f_2 \in \mathcal{F}} \left| R_{\text{coord}}(f_1, f_2) - \hat{R}_{\text{coord}}(f_1, f_2) \right| \leq 2\widehat{\mathcal{R}} \left( l_{\text{coord}}^\circ(\mathcal{F} \times \mathcal{F}) \right) + 3\sqrt{\frac{\log(2/\delta)}{2(n+m)}}
\tag{51}
$$

Regarding the discriminative risk, we decompose the generalization gap by analyzing the expectation and variance terms separately:

$$
\sup_{f_2 \in \mathcal{F}} \left| R_{\text{geo}}(f_2) - \hat{R}_{\text{geo}}(f_2) \right|
$$

$$
\leq \sup_{f_2 \in \mathcal{F}} \left| \mathbb{E}_{\boldsymbol{s} \sim \mathcal{D}_P} \left[ \sigma(f_2(\boldsymbol{s})) \right] - \frac{1}{n} \sum_{\boldsymbol{s} \in S_P} \left[ \sigma(f_2(\boldsymbol{s})) \right] \right| + \sup_{f_2 \in \mathcal{F}} \left| \mathbb{E}_{\boldsymbol{s} \sim \mathcal{D}_U} \left[ \sigma(f_2(\boldsymbol{s})) \right] - \frac{1}{m} \sum_{\boldsymbol{s} \in S_U} \left[ \sigma(f_2(\boldsymbol{s})) \right] \right|
$$

$$
+ \lambda_{\text{com}} \sup_{f_2 \in \mathcal{F}} \left| \mathbb{E}_{\boldsymbol{s} \sim \mathcal{D}_P} \left[ \sigma(f_2(\boldsymbol{s}))^2 \right] - \frac{1}{n} \sum_{\boldsymbol{s} \in S_P} \left[ \sigma(f_2(\boldsymbol{s}))^2 \right] \right| + \lambda_{\text{com}} \sup_{f_2 \in \mathcal{F}} \left| \mathbb{E}_{\boldsymbol{s} \sim \mathcal{D}_P} \left[ \sigma(f_2(\boldsymbol{s})) \right]^2 - \frac{1}{n^2} \left[ \sum_{\boldsymbol{s} \in S_P} \sigma(f_2(\boldsymbol{s})) \right]^2 \right|
$$

$$
= \sup_{f_2 \in \mathcal{F}} \left( \left| \mathbb{E}_{\boldsymbol{s} \sim \mathcal{D}_P} \left[ \sigma(f_2(\boldsymbol{s})) \right] - \frac{1}{n} \sum_{\boldsymbol{s} \in S_P} \left[ \sigma(f_2(\boldsymbol{s})) \right] \right| + \sup_{f_2 \in \mathcal{F}} \left| \mathbb{E}_{\boldsymbol{s} \sim \mathcal{D}_U} \left[ \sigma(f_2(\boldsymbol{s})) \right] - \frac{1}{m} \sum_{\boldsymbol{s} \in S_U} \left[ \sigma(f_2(\boldsymbol{s})) \right] \right| \right.
$$

$$
+ \lambda_{\text{com}} \sup_{f_2 \in \mathcal{F}} \left| \mathbb{E}_{\boldsymbol{s} \sim \mathcal{D}_P} \left[ \sigma(f_2(\boldsymbol{s}))^2 \right] - \frac{1}{n} \sum_{\boldsymbol{s} \in S_P} \left[ \sigma(f_2(\boldsymbol{s}))^2 \right] \right|
$$

$$
\left. + \lambda_{\text{com}} \sup_{f_2 \in \mathcal{F}} \left| \mathbb{E}_{\boldsymbol{s} \sim \mathcal{D}_P} \left[ \sigma(f_2(\boldsymbol{s})) \right] + \frac{1}{n} \sum_{\boldsymbol{s} \in S_P} \left[ \sigma(f_2(\boldsymbol{s})) \right] \right| \cdot \left| \mathbb{E}_{\boldsymbol{s} \sim \mathcal{D}_P} \left[ \sigma(f_2(\boldsymbol{s})) \right] - \frac{1}{n} \sum_{\boldsymbol{s} \in S_P} \left[ \sigma(f_2(\boldsymbol{s})) \right] \right| \right)
$$

$$
\leq \sup_{f_2 \in \mathcal{F}} \left| \mathbb{E}_{\boldsymbol{s} \sim \mathcal{D}_U} \left[ \sigma(f_2(\boldsymbol{s})) \right] - \frac{1}{m} \sum_{\boldsymbol{s} \in S_U} \left[ \sigma(f_2(\boldsymbol{s})) \right] \right| + \lambda_{\text{com}} \sup_{f_2 \in \mathcal{F}} \left| \mathbb{E}_{\boldsymbol{s} \sim \mathcal{D}_P} \left[ \sigma(f_2(\boldsymbol{s}))^2 \right] - \frac{1}{n} \sum_{\boldsymbol{s} \in S_P} \left[ \sigma(f_2(\boldsymbol{s}))^2 \right] \right|
$$

$$
+ (2\lambda_{\text{com}} + 1) \sup_{f_2 \in \mathcal{F}} \left| \mathbb{E}_{\boldsymbol{s} \sim \mathcal{D}_P} \left[ \sigma(f_2(\boldsymbol{s})) \right] - \frac{1}{n} \sum_{\boldsymbol{s} \in S_P} \left[ \sigma(f_2(\boldsymbol{s})) \right] \right|
$$

$$
\leq 2\widehat{\mathcal{R}}_u \left( \sigma \circ \mathcal{F} \right) + 2\lambda_{\text{com}} \widehat{\mathcal{R}}_p \left( \sigma^{2\circ} \mathcal{F} \right) + (4\lambda_{\text{com}} + 2) \widehat{\mathcal{R}}_p \left( \sigma \circ \mathcal{F} \right) + 3\sqrt{\frac{\log(2/\delta)}{2m}} + (9\lambda_{\text{com}} + 3) \sqrt{\frac{\log(2/\delta)}{2n}}
\tag{52}
$$

Applying the contraction results established previously to the decomposed risks, we first derive the refined bound for the PU risk component:

$$
\sup_{f_1, f_2 \in \mathcal{F}} \left| R_{pu}(f_1, f_2) - \hat{R}_{pu}(f_1, f_2) \right|
$$

$$
\leq 2\mathbb{E}_{\boldsymbol{r}} \left[ \sup_{f_1 \in \mathcal{F}} \frac{1}{n} \sum_{i=1}^n r_i f_1(\boldsymbol{s}_i) \right] + 2\mathbb{E}_{\boldsymbol{r}} \left[ \sup_{f_2 \in \mathcal{F}} \frac{1}{n} \sum_{i=1}^n r_i f_2(\boldsymbol{s}_i) \right]
$$

$$+ 2\mathbb{E}_{\boldsymbol{r}}\left[\sup_{f_1 \in \mathcal{F}} \frac{1}{m} \sum_{i=n+1}^{n+m} r_i f_1\left(\boldsymbol{s}_i\right)\right] + 2\mathbb{E}_{\boldsymbol{r}}\left[\sup_{f_2 \in \mathcal{F}} \frac{1}{m} \sum_{i=n+1}^{n+m} r_i f_2\left(\boldsymbol{s}_i\right)\right]$$

$$+ 3A\sqrt{\frac{\log(2/\delta)}{2n}} + 3A\sqrt{\frac{\log(2/\delta)}{2m}} \tag{53}$$

Similarly, by incorporating the contraction on the coordination loss, the bound for the coordination risk is obtained as:

$$\sup_{f_1,f_2 \in \mathcal{F}} \left| R_{\text{coord}}\left(f_1, f_2\right) - \hat{R}_{\text{coord}}\left(f_1, f_2\right) \right|$$

$$\leq 4\mathbb{E}_{\boldsymbol{r}}\left[\sup_{f_1 \in \mathcal{F}} \frac{1}{n+m} \sum_{i=1}^{n+m} r_i f_1\left(\boldsymbol{s}_i\right)\right] \tag{54}$$

$$+ 4\mathbb{E}_{\boldsymbol{r}}\left[\sup_{f_2 \in \mathcal{F}} \frac{1}{n+m} \sum_{i=1}^{n+m} r_i f_2\left(\boldsymbol{s}_i\right)\right] + 3\sqrt{\frac{\log(2/\delta)}{2(n+m)}}$$

Finally, substituting the complexity bounds for the expectation and variance terms of the discriminative risk yields:

$$\sup_{f_2 \in \mathcal{F}} \left| R_{geo}\left(f_2\right) - \hat{R}_{geo}\left(f_2\right) \right|$$

$$\leq \frac{1}{2}\mathbb{E}_{\boldsymbol{r}}\left[\sup_{f_2 \in \mathcal{F}} \frac{1}{m} \sum_{i=n+1}^{n+m} r_i f_2\left(\boldsymbol{s}_i\right)\right]$$

$$+ \left(\frac{43}{27}\lambda_{\text{com}} + \frac{1}{2}\right) \mathbb{E}_{\boldsymbol{r}}\left[\sup_{f_2 \in \mathcal{F}} \frac{1}{n} \sum_{i=1}^{n} r_i f_2\left(\boldsymbol{s}_i\right)\right] \tag{55}$$

$$+ 3\sqrt{\frac{\log(2/\delta)}{2m}} + \left(9\lambda_{\text{com}} + 3\right)\sqrt{\frac{\log(2/\delta)}{2n}}$$

To obtain explicit convergence rates, we instantiate the derived complexity terms using the spectral norm bounds for MLPs (Golowich et al., 2018). Substituting the product of weight matrix norms into Eq. (53), the explicit bound for the PU risk is given by:

$$\sup_{f_1,f_2 \in \mathcal{F}} \left| R_{pu}\left(f_1, f_2\right) - \hat{R}_{pu}\left(f_1, f_2\right) \right|$$

$$\leq 2\frac{B(\sqrt{2l\log 2} + 1)\left(\prod_{i=1}^{l} M_{f_1}^{(i)} + \prod_{i=1}^{l} M_{f_2}^{(i)}\right)}{\sqrt{n}}$$

$$+ 2\frac{B(\sqrt{2l\log 2} + 1)\left(\prod_{i=1}^{l} M_{f_1}^{(i)} + \prod_{i=1}^{l} M_{f_2}^{(i)}\right)}{\sqrt{m}} \tag{56}$$

$$+ 3A\sqrt{\frac{\log(2/\delta)}{2n}} + 3A\sqrt{\frac{\log(2/\delta)}{2m}}$$

By the same token, applying the spectral constraints to the coordination term yields:

$$\sup_{f_1,f_2 \in \mathcal{F}} \left| R_{\text{coord}}\left(f_1, f_2\right) - \hat{R}_{\text{coord}}\left(f_1, f_2\right) \right|$$

$$\leq 4\frac{B(\sqrt{2l\log 2} + 1)\left(\prod_{i=1}^{l} M_{f_1}^{(i)} + \prod_{i=1}^{l} M_{f_2}^{(i)}\right)}{\sqrt{n+m}} + 3\sqrt{\frac{\log(2/\delta)}{2(n+m)}} \tag{57}$$

Lastly, for the discriminative risk which depends solely on the posterior network $f_2$, the bound is instantiated as:

$$
\sup_{f_2 \in \mathcal{F}} \left| R_{\text{geo}}(f_2) - \hat{R}_{\text{geo}}(f_2) \right|
$$
$$
\leq \frac{1}{2} \frac{B(\sqrt{2l \log 2} + 1) \prod_{i=1}^{l} M_{f_2}^{(i)}}{\sqrt{m}}
$$
$$
+ \left( \frac{43}{27} \lambda_{\text{com}} + \frac{1}{2} \right) \frac{B(\sqrt{2l \log 2} + 1) \prod_{i=1}^{l} M_{f_2}^{(i)}}{\sqrt{n}}
$$
$$
+ 3\sqrt{\frac{\log(2/\delta)}{2m}} + (9\lambda_{\text{com}} + 3) \sqrt{\frac{\log(2/\delta)}{2n}}
$$

(58)

Combining these components allows us to bound the total generalization gap:

$$
\left| R_{\text{total}}(f_1, f_2) - \hat{R}_{\text{total}}(f_1, f_2) \right|
$$
$$
\leq \sup_{f_1, f_2 \in \mathcal{F}} \left| R_{\text{total}}(f_1, f_2) - \hat{R}_{\text{total}}(f_1, f_2) \right|
$$
$$
\leq \sup_{f_1, f_2 \in \mathcal{F}} \left| R_{pu}(f_1, f_2) - \hat{R}_{pu}(f_1, f_2) \right|
$$
$$
+ \lambda_{\text{coord}} \sup_{f_1, f_2 \in \mathcal{F}} \left| R_{\text{coord}}(f_1, f_2) - \hat{R}_{\text{coord}}(f_1, f_2) \right|
$$
$$
+ \lambda_{\text{geo}} \sup_{f_2 \in \mathcal{F}} \left| R_{\text{geo}}(f_2) - \hat{R}_{\text{geo}}(f_2) \right|
$$

(59)

Substituting Eq. (56)-(58) into Eq. (59), and grouping the coefficients for $n$, $m$, and $n + m$, the conclusion presented in Eq. (29) holds. □

## B. Implementation Details and Complexity Analysis

### B.1. Network Architectures and Implementation

PUFE parameterizes the two conditional distributions $p(a \mid y, s)$ and $p(y \mid s)$ with two lightweight MLPs: the labeling propensity network $f_1$ and the class posterior network $f_2$. Both networks map the semantic embedding $s \in \mathbb{R}^d$ to a scalar logit, followed by a sigmoid activation.

Despite sharing inputs, the networks adopt deliberately asymmetric capacities that reflect their distinct theoretical roles in the generalization bound (Theorem 4.1), where the complexity terms of $f_1$ and $f_2$ carry different coefficients.
**Labeling Network ($f_1$):** Models instance-dependent labeling propensity, a locally-varying quantity sensitive to semantic typicality. A narrower tapering architecture ($d \rightarrow 384 \rightarrow 192 \rightarrow 96 \rightarrow 48 \rightarrow 1$) constrains the Frobenius norm product $\prod_i M_{f_1}^{(i)}$, keeping its contribution to the bound tight.
**Posterior Network ($f_2$):** Responsible for global seen–unseen discrimination across the full semantic space. A wider bottleneck ($d \rightarrow 640 \rightarrow 320 \rightarrow 160 \rightarrow 1$) provides the representational capacity needed for reliable posterior estimation, whose complexity term in the bound tolerates larger norm products.
Both branches utilize Layer Normalization and LeakyReLU activations, with Dropout applied to enhance robustness. $L_2$ weight decay is applied during training to explicitly enforce the norm bounds assumed in Theorem 4.1. All parameters are optimized jointly via standard backpropagation, as detailed in Algorithm 1.

### B.2. Complexity Analysis

**Time Complexity.** Let $n = |S_P|$ and $m = |S_U|$ denote the sizes of the seen and unlabeled sets, respectively, and $d$ be the semantic dimension. Each iteration performs forward and backward passes through two lightweight MLPs, yielding a per-iteration complexity of $\mathcal{O}((n + m)d)$. Over $T$ iterations, the total complexity is $\mathcal{O}(T(n + m)d)$, scaling linearly with the total sample size.

**Space Complexity.** Memory consumption is dominated by the storage of semantic features and network parameters, resulting in $\mathcal{O}((n + m)d + |\theta|)$. Mini-batch training allows efficient deployment on standard GPUs.

**Convergence.** Empirically, PUFE typically converges within 200–300 iterations, indicating reliable numerical stability.

---

**Algorithm 1** Dual-Network Likelihood Optimization for PUFE

---

**Input:** positives $S_P = \{s_i\}_{i=1}^n$, unlabeled $S_U = \{s_j\}_{j=1}^m$; $\lambda_{\text{coord}}, \lambda_{\text{geo}}, \lambda_{\text{com}}$; $\varepsilon > 0$; iterations $T$.

**Output:** trained parameters $(\theta_1, \theta_2)$ and PU posteriors $\{\sigma(f_2(s))\}_{s \in S_U}$.

1  Initialize two MLPs: labeling-propensity $f_1(\cdot; \theta_1)$ and class-posterior $f_2(\cdot; \theta_2)$

2  **for** $t = 1$ **to** $T$ **do**

3      **Forward.** For all $s \in S_P \cup S_U$: compute network outputs $\sigma(f_1(s; \theta_1))$ and $\sigma(f_2(s; \theta_2))$

4      **PU Empirical Risk** (Eq. (10))

5      $\hat{R}_P \leftarrow -\frac{1}{n} \sum_{s \in S_P} \log\big(\sigma(f_1(s))\,\sigma(f_2(s)) + \varepsilon\big)$

6      $\hat{R}_U \leftarrow -\frac{1}{m} \sum_{s \in S_U} \log\big(1 - \sigma(f_1(s))\,\sigma(f_2(s)) + \varepsilon\big)$

7      $\hat{R}_{pu} \leftarrow \hat{R}_P + \hat{R}_U$

8      **Semantic Coordination** (Eq. (12))

9      $\hat{R}_{\text{coord}} \leftarrow \frac{1}{n+m} \sum_{s \in S_P \cup S_U} \big(\sigma(f_1(s)) - \sigma(f_2(s))\big)^2$

10     **Discriminative Geometry Regularization** (Eqs. (14)–(16))

11     $\bar{\mu}_P \leftarrow \frac{1}{n} \sum_{s \in S_P} \sigma(f_2(s)), \bar{\mu}_U \leftarrow \frac{1}{m} \sum_{s \in S_U} \sigma(f_2(s))$

12     $\hat{R}_{\text{sep}} \leftarrow -(\bar{\mu}_P - \bar{\mu}_U)$

13     $\hat{R}_{\text{com}} \leftarrow \frac{1}{n} \sum_{s \in S_P} \big(\sigma(f_2(s)) - \bar{\mu}_P\big)^2$

14     $\hat{R}_{\text{geo}} \leftarrow \hat{R}_{\text{sep}} + \lambda_{\text{com}}\hat{R}_{\text{com}}$

15     **Total Objective** (Eq. (17))

16     $\hat{R}_{\text{total}} \leftarrow \hat{R}_{pu} + \lambda_{\text{coord}}\hat{R}_{\text{coord}} + \lambda_{\text{geo}}\hat{R}_{\text{geo}}$

17     **Update.** Apply one step of any first-order optimizer to minimize $\hat{R}_{\text{total}}$:

18     $(\theta_1, \theta_2) \leftarrow \text{OPTSTEP}\big((\theta_1, \theta_2), \nabla \hat{R}_{\text{total}}\big)$

19 **end**

20 **return** $(\theta_1, \theta_2)$ and $\{\sigma(f_2(s))\}_{s \in S_U}$

---

## C. PU-Guided Adaptive Semantic Calibration

This appendix supplements Section 3.2 by providing the complete algorithmic pseudocode, additional details on confidence-aware inference, and the rigorous proof of the calibration Guarantee (Theorem 4.2).

### C.1. Inference with Confidence Calibration

Following the semantic prototype calibration, we design an auxiliary confidence-guided adjustment strategy to further refine the predictions during inference. We first compute the base similarities $S$ between the test instance $x$ and the calibrated prototypes $\mathcal{A}^{(1)}$ (obtained in Section 3.2). To dynamically fine-tune these similarities, we employ the class posterior $f_2(s)$ learned by the PU classifier.

We map the raw PU output to a soft confidence weight $w = \tanh(f_2(s)) \in (-1, 1)$. This weight is then applied to modulate the similarity scores for seen classes ($\mathcal{C}^s$) and unseen classes ($\mathcal{C}^u$) as follows:

$$s_c \leftarrow \begin{cases} s_c \cdot (1 + \alpha w), & \text{if } c \in \mathcal{C}^s \\ s_c \cdot (1 - \alpha w), & \text{if } c \in \mathcal{C}^u \end{cases} \tag{60}$$

where $s_c$ denotes the similarity score for class $c$, and $\alpha$ is a scaling factor set to 0.1. This mechanism works intuitively: instances with high seen-confidence ($w > 0$) receive a boost in seen-class similarities and a penalty in unseen-class ones, whereas instances showing unseen tendencies ($w < 0$) are adjusted conversely to favor unseen categories.

---

**Algorithm 2** PU-Guided Adaptive Semantic Calibration

---

**Input:** $S_U = \{\boldsymbol{q}_j\}_{j=1}^m$; PU posteriors $\{p_j\}$; prototypes $\mathcal{A}^{(0)} = \{\boldsymbol{a}_c^{(0)}\}_{c \in \mathcal{C}}$; scaling factors $\kappa_+, \kappa_-$

**Output:** Calibrated prototypes $\mathcal{A}^{(1)} = \{\boldsymbol{a}_c^{(1)}\}_{c \in \mathcal{C}}$

21 $\mu_p \leftarrow \frac{1}{m} \sum_{j=1}^m p_j, \quad \sigma_p \leftarrow \sqrt{\frac{1}{m} \sum_{j=1}^m (p_j - \mu_p)^2}$

22 $\tau_+ \leftarrow \mu_p + \kappa_+ \sigma_p, \quad \tau_- \leftarrow \mu_p - \kappa_- \sigma_p$

23 Initialize supports $E_c \leftarrow \emptyset$ for all $c \in \mathcal{C}$

24 **for** $j = 1$ **to** $m$ **do**

25     **for** *each class* $c \in \mathcal{C}$ **do**

26         $Z_{j,c} \leftarrow \text{Sim}(\boldsymbol{q}_j, \boldsymbol{a}_c^{(0)})$     // Sim($\cdot$) is dot product or cosine similarity depending on baselines

27     **end**

28 **end**

29 **for** $j = 1$ **to** $m$ **do**

30     **if** $p_j > \tau_+$ **then**

31         $\hat{c}_j \leftarrow \arg\max_{c \in \mathcal{C}_{seen}} Z_{j,c}$

32         $E_{\hat{c}_j} \leftarrow E_{\hat{c}_j} \cup \{\boldsymbol{q}_j\}$

33     **end**

34     **else if** $p_j < \tau_-$ **then**

35         $\hat{c}_j \leftarrow \arg\max_{c \in \mathcal{C}_{unseen}} Z_{j,c}$

36         $E_{\hat{c}_j} \leftarrow E_{\hat{c}_j} \cup \{\boldsymbol{q}_j\}$

37     **end**

38 **end**

39 **for** *each class* $c \in \mathcal{C}$ **do**

40     **if** $|E_c| \geq N_{\min}$ **then**

41         $\boldsymbol{\mu}_c \leftarrow \frac{1}{|E_c|} \sum_{\boldsymbol{q} \in E_c} \boldsymbol{q}; \quad \bar{\boldsymbol{\mu}}_c \leftarrow \boldsymbol{\mu}_c / \|\boldsymbol{\mu}_c\|_2$

42         $\rho \leftarrow \boldsymbol{a}_c^{(0)T} \bar{\boldsymbol{\mu}}_c; \quad \alpha_c \leftarrow \max(\alpha_{\min}, \min(\rho, \alpha_{\max}))$

43         $\boldsymbol{u}_c \leftarrow \alpha_c \boldsymbol{a}_c^{(0)} + (1 - \alpha_c) \bar{\boldsymbol{\mu}}_c$

44         $\boldsymbol{a}_c^{(1)} \leftarrow \boldsymbol{u}_c / \|\boldsymbol{u}_c\|_2$

45     **end**

46     **else**

47         $\boldsymbol{a}_c^{(1)} \leftarrow \boldsymbol{a}_c^{(0)}$

48     **end**

49 **end**

50 **return** $\mathcal{A}^{(1)}$

---

### C.2. Theoretical Guarantee of Calibration

Let the ground-truth semantic centroid of category $c$ be $\boldsymbol{\mu}_c^* = \frac{\mathbb{E}_{\boldsymbol{q} \sim \mathcal{S}_c}(\boldsymbol{q})}{\|\mathbb{E}_{\boldsymbol{q} \sim \mathcal{S}_c}(\boldsymbol{q})\|_2}$, where $\mathcal{S}_c$ is the semantic distribution of category $c$. Denoting the pseudo-centroid of $E_c$ as $\bar{\boldsymbol{\mu}}_c$, we have:

$$\bar{\boldsymbol{\mu}}_c = \frac{\frac{1}{|E_c|} \sum_{\boldsymbol{q} \in E_c} \boldsymbol{q}}{\left\| \frac{1}{|E_c|} \sum_{\boldsymbol{q} \in E_c} \boldsymbol{q} \right\|_2} \tag{61}$$

**Lemma C.1.** $\bar{\boldsymbol{\mu}}_c$ *yields a better estimation for* $\boldsymbol{\mu}_c^*$ *than the original semantic prototype* $\boldsymbol{a}_c^{(0)}$, *i.e.,* $\text{sim}\left(\bar{\boldsymbol{\mu}}_c, \boldsymbol{\mu}_c^*\right) > \text{sim}\left(\boldsymbol{a}_c^{(0)}, \boldsymbol{\mu}_c^*\right)$, *where* sim *denotes the baseline-dependent metric.*

*Proof.* Let $\varepsilon_s = \left\| \boldsymbol{a}_c^{(0)} - \boldsymbol{\mu}_c^* \right\|_2$ denote the semantic bias between the original prototype $\boldsymbol{a}_c^{(0)}$ and the ground-truth semantic centroid $\boldsymbol{\mu}_c^*$, and $\varepsilon_e = \|\bar{\boldsymbol{\mu}}_c - \boldsymbol{\mu}_c^*\|_2$ denote the estimation error of the pseudo-centroid $\bar{\boldsymbol{\mu}}_c$ derived from the high-confidence support set for category $c$. We assume the PU classifier provides a sufficiently discriminative selection such that the estimation bias is strictly bounded by the semantic bias:

$$\varepsilon_e < \varepsilon_s \tag{62}$$

Thus, we have:

$$\|\bar{\boldsymbol{\mu}}_c - \boldsymbol{\mu}_c^*\|_2 < \left\| \boldsymbol{a}_c^{(0)} - \boldsymbol{\mu}_c^* \right\|_2 \tag{63}$$

Considering $\|\bar{\boldsymbol{\mu}}_c\|_2 = \left\|\boldsymbol{a}_c^{(0)}\right\|_2 = 1$, Eq. (63) is rewritten into:

$$\bar{\boldsymbol{\mu}}_c^T \boldsymbol{\mu}_c^* > \boldsymbol{a}_c^{(0)^T} \boldsymbol{\mu}_c^* \tag{64}$$

Also, we have:

$$\cos\left(\bar{\boldsymbol{\mu}}_c, \boldsymbol{\mu}_c^*\right) = \frac{\bar{\boldsymbol{\mu}}_c^T \boldsymbol{\mu}_c^*}{\|\bar{\boldsymbol{\mu}}_c\|_2 \|\boldsymbol{\mu}_c^*\|_2} = \frac{\bar{\boldsymbol{\mu}}_c^T \boldsymbol{\mu}_c^*}{\|\boldsymbol{\mu}_c^*\|_2} \tag{65}$$

$$\cos\left(\boldsymbol{a}_c^{(0)}, \boldsymbol{\mu}_c^*\right) = \frac{\boldsymbol{a}_c^{(0)^T} \boldsymbol{\mu}_c^*}{\left\|\boldsymbol{a}_c^{(0)}\right\|_2 \|\boldsymbol{\mu}_c^*\|_2} = \frac{\boldsymbol{a}_c^{(0)^T} \boldsymbol{\mu}_c^*}{\|\boldsymbol{\mu}_c^*\|_2} \tag{66}$$

So,

$$\cos\left(\bar{\boldsymbol{\mu}}_c, \boldsymbol{\mu}_c^*\right) > \cos\left(\boldsymbol{a}_c^{(0)}, \boldsymbol{\mu}_c^*\right). \tag{67}$$

Therefore, Eqs. (64) and (67) jointly imply $\mathrm{sim}(\bar{\boldsymbol{\mu}}_c, \boldsymbol{\mu}_c^*) > \mathrm{sim}(\boldsymbol{a}_c^{(0)}, \boldsymbol{\mu}_c^*)$. $\qquad\square$

**Proof of Theorem 4.2.** *Statement: If the calibrated prototype differs from the original (i.e., $\alpha_c < 1$), then $\mathrm{sim}(\boldsymbol{a}_c^{(1)}, \boldsymbol{\mu}_c^*) > \mathrm{sim}(\boldsymbol{a}_c^{(0)}, \boldsymbol{\mu}_c^*)$, where $\mathrm{sim}$ denotes the baseline-dependent metric.*

*Proof.* When $\boldsymbol{a}_c^{(1)} \neq \boldsymbol{a}_c^{(0)}$, we have $\alpha_c < 1$. First, expanding the projection of the unnormalized update vector onto the ground truth:

$$\left(\alpha_c \boldsymbol{a}_c^{(0)} + (1 - \alpha_c)\bar{\boldsymbol{\mu}}_c\right)^T \boldsymbol{\mu}_c^* = \alpha_c \boldsymbol{a}_c^{(0)^T} \boldsymbol{\mu}_c^* + (1 - \alpha_c)\bar{\boldsymbol{\mu}}_c^T \boldsymbol{\mu}_c^*. \tag{68}$$

Using Lemma C.1 ($\bar{\boldsymbol{\mu}}_c^T \boldsymbol{\mu}_c^* > \boldsymbol{a}_c^{(0)^T} \boldsymbol{\mu}_c^*$), we have:

$$\left(\alpha_c \boldsymbol{a}_c^{(0)} + (1 - \alpha_c)\bar{\boldsymbol{\mu}}_c\right)^T \boldsymbol{\mu}_c^* > \alpha_c \boldsymbol{a}_c^{(0)^T} \boldsymbol{\mu}_c^* + (1 - \alpha_c)\boldsymbol{a}_c^{(0)^T} \boldsymbol{\mu}_c^* = \boldsymbol{a}_c^{(0)^T} \boldsymbol{\mu}_c^*. \tag{69}$$

Next, bounding the norm using the triangle inequality:

$$\left\|\alpha_c \boldsymbol{a}_c^{(0)} + (1 - \alpha_c)\bar{\boldsymbol{\mu}}_c\right\|_2 \leq \alpha_c \left\|\boldsymbol{a}_c^{(0)}\right\|_2 + (1 - \alpha_c)\|\bar{\boldsymbol{\mu}}_c\|_2 = 1. \tag{70}$$

Now, analyzing the normalized projection for $\boldsymbol{a}_c^{(1)}$:

$$\boldsymbol{a}_c^{(1)^T} \boldsymbol{\mu}_c^* = \frac{1}{\left\|\alpha_c \boldsymbol{a}_c^{(0)} + (1 - \alpha_c)\bar{\boldsymbol{\mu}}_c\right\|_2} \left(\alpha_c \boldsymbol{a}_c^{(0)} + (1 - \alpha_c)\bar{\boldsymbol{\mu}}_c\right)^T \boldsymbol{\mu}_c^*. \tag{71}$$

Combining the numerator inequality and the norm bound (denominator $\leq 1$), we obtain:

$$\boldsymbol{a}_c^{(1)^T} \boldsymbol{\mu}_c^* > \frac{1}{\left\|\alpha_c \boldsymbol{a}_c^{(0)} + (1 - \alpha_c)\bar{\boldsymbol{\mu}}_c\right\|_2} \boldsymbol{a}_c^{(0)^T} \boldsymbol{\mu}_c^* \geq \boldsymbol{a}_c^{(0)^T} \boldsymbol{\mu}_c^*. \tag{72}$$

Finally, by the definition of cosine similarity:

$$\cos(\boldsymbol{a}_c^{(1)}, \boldsymbol{\mu}_c^*) = \frac{\boldsymbol{a}_c^{(1)^T} \boldsymbol{\mu}_c^*}{\left\|\boldsymbol{a}_c^{(1)}\right\|_2 \|\boldsymbol{\mu}_c^*\|_2} = \boldsymbol{a}_c^{(1)^T} \boldsymbol{\mu}_c^*, \tag{73}$$

$$\cos(\boldsymbol{a}_c^{(0)}, \boldsymbol{\mu}_c^*) = \frac{\boldsymbol{a}_c^{(0)^T} \boldsymbol{\mu}_c^*}{\left\|\boldsymbol{a}_c^{(0)}\right\|_2 \|\boldsymbol{\mu}_c^*\|_2} = \boldsymbol{a}_c^{(0)^T} \boldsymbol{\mu}_c^*. \tag{74}$$

Thus, we conclude:

$$\cos(\boldsymbol{a}_c^{(1)}, \boldsymbol{\mu}_c^*) > \cos(\boldsymbol{a}_c^{(0)}, \boldsymbol{\mu}_c^*). \tag{75}$$

This confirms that the calibrated prototype $\boldsymbol{a}_c^{(1)}$ achieves a stricter alignment with the ground truth than the original $\boldsymbol{a}_c^{(0)}$ regardless of whether the dot product (Eq. 72) or cosine similarity (Eq. 75) is employed as the metric $\mathrm{sim}(\cdot, \cdot)$. $\qquad\square$

# D. Experimental Setup

## D.1. Datasets and Evaluation Protocol

To systematically evaluate the robustness of PUFE across diverse semantic granularities and distribution structures, we adopt three widely used zero-shot learning benchmarks: CUB, SUN, and AWA2. All experiments follow the standard GZSL data split protocol proposed by Xian et al. (Xian et al., 2018a).

Detailed statistics and characteristics are summarized in Table 5. Among them, **CUB** challenges the model with fine-grained distinctions; **SUN** serves as a stress test for heterogeneity; and **AWA2** provides a robust benchmark for generalizability.

*Table 5.* Statistics and Characteristics of GZSL Datasets.

| Dataset | Classes | Split (S/U) | Images | Dim | Characteristics |
|---------|---------|-------------|--------|-----|-----------------|
| **CUB** | 200 | 150 / 50 | 11,788 | 312 | Demands the discrimination of subtle semantic nuances amidst highly complex visual backgrounds. |
| **SUN** | 717 | 645 / 72 | 14,340 | 102 | A large-scale stress test defined by extreme inter-class diversity and sparse attribute annotations. |
| **AWA2** | 50 | 40 / 10 | 37,322 | 85 | Features compact semantic clusters and clear boundaries, serving as a standard reference for generalizability. |

## D.2. Baseline Models

To assess the adaptability of PUFE as a plug-and-play module, we integrate it into seven representative SOTA models spanning various architectural paradigms.

Table 6 compares their core mechanisms. Despite their architectural diversity, all baselines benefit from PUFE's unified ability to explicitly handle label incompleteness.

*Table 6.* Summary of Representative Baseline Models.

| Model | Mechanism | Backbone | Distinction |
|-------|-----------|----------|-------------|
| **TransZero** | Attribute-Guided Attn. | Transformer | Localizes informative visual regions via attribute-guided attention to enhance fine-grained discrimination. |
| **MSDN** | Semantic Distillation | Attention-based | Distills intrinsic cross-modal correlations to construct robust, semantic-consistent representations. |
| **IFMVZSL** | Multi-view Fusion | Multi-view | Leverages complementary feature views to enforce prediction consistency via collaborative regularization. |
| **PSVMA** | Progressive Alignment | Transformer | Counteracts seen-class bias by progressively aligning visual tokens with semantic prototypes. |
| **ZSLViT** | Semantic-Guided Pruning | Vision Transformer | Purifies visual embeddings by pruning irrelevant tokens layer-by-layer via semantic-visual interaction. |
| **ZeroMamba** | Vision Mamba + Semantic Fusion | Vision Mamba | Enhances visual-semantic interaction through semantic-aware local projection and global representation fusion within a parameter-efficient Mamba-based framework. |
| **TPR** | Dual-space Alignment | CLIP-based VLM | Preserves semantic topology across seen and unseen classes via topology-preserving reservoirs and dual-space visual-textual alignment. |

# E. Detailed Experimental Analysis and Robustness Verification

## E.1. Statistical Stability Analysis

To evaluate the optimization stability of PUFE, we report the mean and standard deviation of performance metrics over 5 independent runs in Table 7. The results indicate that PUFE exhibits low variance across diverse baselines and datasets (e.g., standard deviations for $H$ are consistently below 0.7%), confirming that the stochastic gradient descent optimization for the dual-head network converges to a stable solution.

*Table 7.* Performance statistics (Mean $\pm$ Std) of PUFE over 5 independent runs.

| Baseline | Dataset | PU Acc (%) | U (%) | S (%) | H (%) |
|---|---|---|---|---|---|
| **TransZero** | **CUB** | $86.3 \pm 0.2$ | $73.7 \pm 0.62$ | $76.8 \pm 0.51$ | $75.2 \pm 0.53$ |
| | **SUN** | $69.4 \pm 0.5$ | $46.6 \pm 0.65$ | $39.1 \pm 0.58$ | $42.5 \pm 0.60$ |
| | **AWA2** | $98.2 \pm 0.1$ | $71.0 \pm 0.42$ | $93.2 \pm 0.35$ | $80.6 \pm 0.38$ |
| **MSDN** | **CUB** | $90.9 \pm 0.3$ | $78.4 \pm 0.55$ | $74.3 \pm 0.61$ | $76.3 \pm 0.57$ |
| | **SUN** | $68.7 \pm 0.4$ | $52.1 \pm 0.72$ | $37.5 \pm 0.65$ | $43.6 \pm 0.68$ |
| | **AWA2** | $95.5 \pm 0.2$ | $65.4 \pm 0.45$ | $84.1 \pm 0.38$ | $73.6 \pm 0.41$ |
| **IFMVZSL** | **CUB** | $89.6 \pm 0.4$ | $67.6 \pm 0.58$ | $70.4 \pm 0.52$ | $68.9 \pm 0.54$ |
| | **SUN** | $70.3 \pm 0.5$ | $66.7 \pm 0.75$ | $31.6 \pm 0.62$ | $42.8 \pm 0.66$ |
| | **AWA2** | $96.1 \pm 0.2$ | $89.2 \pm 0.35$ | $75.9 \pm 0.44$ | $82.0 \pm 0.39$ |
| **PSVMA** | **CUB** | $96.9 \pm 0.2$ | $83.8 \pm 0.45$ | $84.5 \pm 0.48$ | $84.1 \pm 0.46$ |
| | **SUN** | $70.1 \pm 0.6$ | $61.9 \pm 0.70$ | $44.6 \pm 0.65$ | $51.8 \pm 0.67$ |
| | **AWA2** | $97.2 \pm 0.7$ | $80.4 \pm 0.15$ | $93.8 \pm 0.28$ | $86.6 \pm 0.22$ |
| **ZSLViT** | **CUB** | $90.2 \pm 0.3$ | $79.1 \pm 0.52$ | $77.9 \pm 0.55$ | $78.5 \pm 0.53$ |
| | **SUN** | $70.8 \pm 0.5$ | $57.0 \pm 0.68$ | $42.1 \pm 0.63$ | $48.4 \pm 0.65$ |
| | **AWA2** | $95.0 \pm 0.2$ | $74.8 \pm 0.41$ | $95.2 \pm 0.25$ | $83.8 \pm 0.32$ |

## E.2. Impact Analysis of PU classifier on Calibration

*Table 8.* Detailed performance statistics under controlled PU degradation.

**(a) CUB Dataset (Baseline: TransZero + PUFE)**

| $\alpha$ | PU Acc | Seen (S) (%) | Unseen (U) (%) | Harmonic (H) (%) |
|---|---|---|---|---|
| 0.00 | $0.863 \pm 0.002$ | $76.84 \pm 0.51$ | $73.69 \pm 0.62$ | $75.23 \pm 0.53$ |
| 0.10 | $0.829 \pm 0.007$ | $76.48 \pm 0.86$ | $72.66 \pm 0.54$ | $74.52 \pm 0.47$ |
| 0.20 | $0.794 \pm 0.011$ | $74.96 \pm 1.42$ | $68.31 \pm 0.79$ | $71.48 \pm 0.62$ |
| 0.30 | $0.762 \pm 0.008$ | $75.53 \pm 0.94$ | $70.22 \pm 0.41$ | $72.78 \pm 0.54$ |
| 0.40 | $0.718 \pm 0.004$ | $74.52 \pm 0.71$ | $65.93 \pm 1.06$ | $69.96 \pm 0.66$ |

**(b) AWA2 Dataset (Baseline: PSVMA + PUFE)**

| $\alpha$ | PU Acc | Seen (S) (%) | Unseen (U) (%) | Harmonic (H) (%) |
|---|---|---|---|---|
| 0.00 | $0.972 \pm 0.007$ | $93.82 \pm 0.28$ | $80.43 \pm 0.15$ | $86.63 \pm 0.22$ |
| 0.10 | $0.944 \pm 0.004$ | $93.04 \pm 0.64$ | $79.97 \pm 0.46$ | $86.01 \pm 0.36$ |
| 0.20 | $0.916 \pm 0.005$ | $90.48 \pm 0.87$ | $79.21 \pm 0.71$ | $84.47 \pm 0.54$ |
| 0.30 | $0.878 \pm 0.009$ | $91.53 \pm 0.91$ | $79.37 \pm 1.22$ | $85.02 \pm 0.61$ |
| 0.40 | $0.832 \pm 0.023$ | $87.49 \pm 1.52$ | $78.61 \pm 0.84$ | $82.81 \pm 1.26$ |

$\alpha$ denotes the noise injection rate. Metrics reported are mean $\pm$ standard deviation.

To strictly validate the dependency of our proposed semantic calibration mechanism on the quality of PU classifier, we conducted a controlled degradation experiment. The primary objective is to verify whether the performance gains in GZSL

are causally linked to the accuracy of the PU classifier.

We deliberately degraded the discriminative capability of the PU classifier by injecting controlled label noise during the PU learning phase. Specifically, we introduced a noise ratio parameter $\alpha \in \{0.0, 0.1, 0.2, 0.3, 0.4\}$, representing the proportion of positive labels randomly flipped to negative. This setup allows us to observe the variations in GZSL performance (measured by the harmonic mean $H$) as the PU classification accuracy (PU Acc) declines, while keeping the baseline GZSL model structure and parameters unchanged. Here, PSVMA on AWA2 and TransZero on CUB were taken as two examples.

Table 8 details the quantitative statistics of this analysis. We performed a Pearson correlation analysis based on this data to quantify the relationship between PU accuracy and the $H$ score. On the fine-grained CUB dataset, the degradation in PU accuracy leads to an overall declining trend in GZSL performance. The statistical analysis reveals a strong positive correlation ($r = 0.909$, $p = 0.032$), indicating that in scenarios where inter-class margins are narrow, a high-precision classifier for seen and unseen classes is critical for effective calibration. Similarly, on the coarse-grained AWA2 dataset, the correlation remains significant ($r = 0.927$, $p = 0.023$). This dependency is further corroborated by a trial-level analysis across all independent experimental runs ($N = 25$), which yields a high statistical significance ($p < 0.001$) for both datasets, confirming that our findings hold despite the limited sample size. However, the performance degradation is relatively mitigated compared to CUB, suggesting that AWA2 is slightly more robust to noise due to its distinct semantic cluster separation.

Figure 8 visualizes this linear relationship. It confirms that high-quality PU posteriors serve as reliable "semantic saliency" signals. They effectively guide the PUFE module to select discriminative representations for calibrating category prototypes, thereby directly contributing to the final generalization performance.

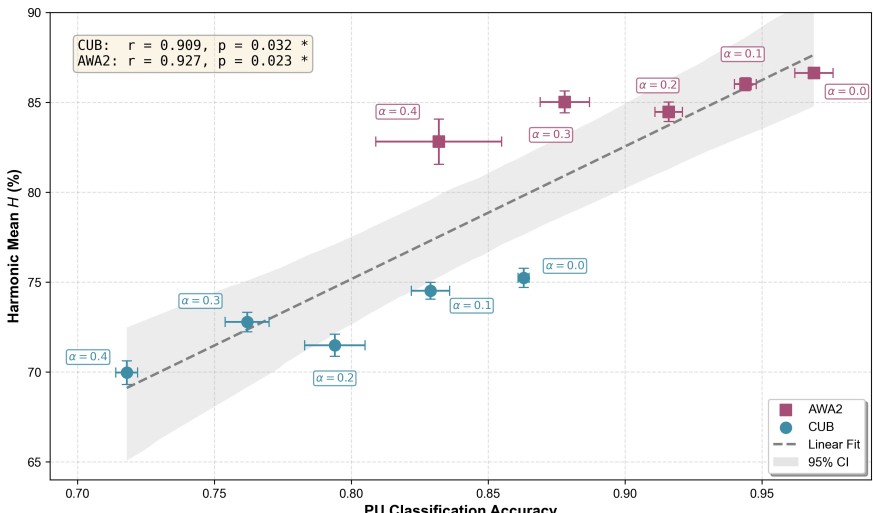

*Figure 8.* Impact of PU learning quality on zero-shot semantic calibration performance. The scatter plot illustrates the positive correlation between PU classification accuracy and the GZSL Harmonic mean ($H$). Blue circles denote results on CUB (Baseline: TransZero), and red squares denote results on AWA2 (Baseline: PSVMA). Error bars represent standard deviations over 5 independent runs.

