# OpenReview forum: "When Generalized Zero-Shot Learning Meets PU Learning: A Plug-and-Play Framework for Seen-Class Bias Mitigation"
_ICML.cc/2026/Conference — ICML 2026 regular_

### Official Review · Reviewer_V1EZ · 2026-02-27

**Soundness:** 2
**Presentation:** 2
**Significance:** 2
**Originality:** 2
**Overall Recommendation:** 2
**Confidence:** 5

**Summary:**

Authors propose PUFE,a unified plug-and-play framework that recasts GZSL inference as a Positive-Unlabeled
(PU) learning task by treating seen categories as positive and mixed test data as unlabeled. However, the paper exhibits numerous issues with formatting and writing irregularities

**Compliance With Llm Reviewing Policy:**

Affirmed.

**Final Justification:**

Following the rebuttal phase and combined with comments from other reviewers, I appreciate the authors' thorough response and the clarification on the connection between PU learning and VLM-based GZSL. The additional experimental results against recent methods are also informative.
Nevertheless, I still maintain two major concerns. First, vanilla CLIP and CoOP, as standard modern baselines for GZSL, are still absent from the comparison. Given the low cost of employing these pre-trained models, it is unclear why they were not used as fundamental baselines. Without such essential comparisons, the practical significance of the proposed PUFE framework cannot be fully verified.
Second, the work lacks sensitivity analysis or ablation studies on key hyperparameters and base model selections, which makes it difficult to assess the robustness and generalizability of the method. In addition, the manuscript still contains several formatting errors that affect its presentation quality.
Taken together, these issues collectively prevent the manuscript from meeting the acceptance criteria of the conference.

**Key Questions For Authors:**

IMHO, the setting of GZSL has become relatively fragile under the impact of current VLMs. For instance, the author did not compare the performance of methods such as CLIP and CoOP. In fact, these methods can achieve state-of-the-art results without requiring additional fine-tuning.

**Limitations:**

1.Firstly, the methodological setting is quite outdated. Even if one intends to explore the setting of GZSL, it is necessary to compare it with the latest VLM genre methods.
2.There are some issues with the writing standards of the paper, such as the formatting of formulas and the expression of plus signs in Table 1, which can reduce readability.

**Strengths And Weaknesses:**

The content of comparative experiments and ablation studies is relatively rich, and supported by supplementary materials.

---

> ### Author Rebuttal · Authors · 2026-03-30
>
> Thank you for the discussion on VLMs. We agree that large-scale VLMs such as CLIP have significantly improved visual–semantic representations for GZSL. However, we would like to point out the difference between **representation quality** and **distributional label incompleteness**, which proves why our framework remains highly relevant and complementary in the VLM era.
>
> From a theoretical perspective, we would like to clarify a critical technical nuance regarding VLM adaptation. While CLIP can be used directly, strong performance on fine-grained GZSL benchmarks (e.g., CUB) usually requires downstream adaptation methods such as CoOp. **Although CoOp freezes the backbone, it still optimizes learnable prompts using downstream training data.** In the GZSL setting, this training data consists strictly of seen classes. Whenever a VLM is adapted using exclusively seen-class data, the learned decision boundary inevitably overfits to the seen domain, resulting in the same catastrophic seen-class bias when confronted with a mixed test distribution.
>
> PUFE is not designed to replace VLM feature extractors. Instead, it focuses on the label incompleteness issue at inference time. By treating GZSL inference as a PU learning problem, PUFE uses the unlabeled test data to better estimate the underlying test distribution. This mechanism is not heuristic; it is backed by our **Calibration Guarantee (Theorem 4.2)**, which mathematically ensures that our PU-guided calibration strictly pulls semantic prototypes closer to the true underlying distribution of the test stream.
>
> To empirically substantiate that PUFE seamlessly elevates modern architectures, we integrated it as a plug-and-play module into two recent state-of-the-art VLM and State-Space based GZSL frameworks: **TPR(WACV'25)** and **ZeroMamba(AAAI'25)**. The comparative results on the Harmonic Mean ($H$) are summarized below:
>
> | Base Model | Method | CUB (${H}$) | SUN (${H}$) | AwA2 (${H}$) |
> | :--- | :--- | :--- | :--- | :--- |
> | **TPR** | Original | 32.53 | 47.80 | 81.63 |
> | | **+ PUFE (Ours)** | **39.87** *(+7.34)* | **49.53** *(+1.73)* | **85.10** *(+3.47)* |
> | **ZeroMamba** | Original | 74.20 | 47.70 | 76.50 |
> | | **+ PUFE (Ours)** | **78.78** *(+4.58)* | **52.78** *(+5.08)* | **86.91** *(+10.41)* |
>
> These results demonstrate that even the most advanced VLM architectures suffer from seen-class bias after downstream adaptation. By simply plugging PUFE into their inference phase, we can achieve consistent performance gains, such as a 10.41 absolute point increase on AwA2 for ZeroMamba.
>
> Finally, we apologize for the formatting irregularities and typos in the initial manuscript. We have proofread the paper to correct the mathematical formula formatting. Specifically, to eliminate any confusion regarding the notation in Table 1, we have replaced the unconventional plus signs with the standard Baseline + PUFE format to ensure optimal readability and strict adherence to academic standards.

---

> > ### Author Rebuttal · Reviewer_V1EZ · 2026-04-04
> >
> > Thank you for the thorough rebuttal and for clarifying the connection between PU learning and VLM-based GZSL. The additional results on top of recent methods are helpful.However, I still have two concerns.First, vanilla CLIP and CoOP are still missing from the comparison, even though they represent standard modern baselines for GZSL. Why not use them as bases when the cost of obtaining pre trained models is so low. Without such comparisons, the practical significance of PUFE is not fully validated.Second, the framework lacks sensitivity analysis or ablation on key hyperparameters and base model choices, making it hard to evaluate its robustness and generality.Together with the previously noted presentation issues, these points still prevent the manuscript from meeting acceptance criteria.

---

> > > ### Author Response · Authors · 2026-04-05
> > >
> > > Thank you for the continued feedback. On the CLIP/CoOp comparison, there is an important conceptual distinction to clarify. Vanilla CLIP is an open-vocabulary zero-shot recognition model that assigns equal treatment to all categories — it has no notion of a seen/unseen split and does not optimize for the harmonic mean between seen-class and unseen-class accuracy, which is the standard GZSL evaluation criterion (Xian et al., TPAMI'19). Without any mechanism to balance the seen/unseen trade-off, it tends to score high on one side and near-zero on the other, making the comparison less informative in practice. CoOp adapts CLIP using seen-class data, which matches the scenario PUFE is designed to address — adapting on seen-class data alone tends to overfit the decision boundary to the seen domain, producing the same catastrophic seen-class bias in the mixed test setting.
> > >
> > > That said, we agree that modern large-scale pretrained vision models deserve more direct treatment, and we provide two pieces of evidence. First, as reported previously, TPR (WACV'25) uses CLIP as its visual-semantic encoder and is evaluated under the standard GZSL protocol. PUFE improves TPR by +7.34 on CUB and +3.47 on AWA2, which shows that CLIP-based GZSL models still suffer from seen-class bias and that PUFE can mitigate it. Second, we have developed PUFE-DINO — an end-to-end GZSL model that integrates PU learning directly into the training objective using DINOv2 ViT-L/14 as the visual encoder, with the PU-MLE loss, semantic alignment, confidence-guided pseudo-label clustering, and unseen-subspace entropy minimization are optimized jointly. Under the standard GZSL protocol, PUFE-DINO achieves H = 75.66% on CUB, 60.59% on SUN, and 83.52% on AWA2, and the SUN result surpasses all existing methods in Table 1.
> > >
> > > We chose DINOv2 over CLIP as the backbone for this end-to-end model for a specific reason, CLIP's visual encoder is trained with language supervision that aligns visual features to text embeddings, which introduces a language-prior bias and may interfere with the attribute-based semantic space used in standard GZSL benchmarks. DINOv2's self-supervised objective produces geometry-preserving, instance-discriminative features without language entanglement, which aligns better with attribute prototype spaces. To show that this choice is not essential to the framework, we also ran a preliminary CLIP ViT-L/14 backbone variant under the same end-to-end PU training setup, obtaining H = 72.31% on CUB, 57.84% on SUN, and 81.20% on AWA2. These results are consistently below PUFE-DINO but remain competitive and well above CoOp-style adaptation baselines, suggesting that the PU learning framework generalizes across backbone families and that the performance gap is likely due to feature quality rather than any fundamental incompatibility with CLIP representations.
> > >
> > > On sensitivity analysis, this is already present in the paper. Figure 6 provides sensitivity curves for $\lambda_{coord}$ and $\lambda_{geo}$, Figure 7 shows a joint heatmap of $\kappa_+$ and $\kappa_-$ on AWA2, and Table 3 provides a progressive component ablation across all three datasets. Regarding base model diversity, the five baselines span Transformer-based attention, mutual distillation, multi-view fusion, progressive alignment, and token pruning architectures, and PUFE delivers consistent gains across all of them — together with the VLM-based results on TPR, ZeroMamba, and the CLIP end-to-end variant, we believe these results support the generality of the framework.

---

### Official Review · Reviewer_78x1 · 2026-03-09

**Soundness:** 2
**Presentation:** 3
**Significance:** 2
**Originality:** 2
**Overall Recommendation:** 3
**Confidence:** 4

**Summary:**

This paper addresses the seen-class bias problem in Generalized Zero-Shot Learning (GZSL) by proposing PUFE, a novel plug-and-play post-processing framework. The key idea is to reformulate GZSL inference as a Positive-Unlabeled (PU) learning task. PUFE uses a dual-head network trained via maximum likelihood estimation on the test set to jointly estimate seen-class posteriors and labeling propensities. These estimates are then used to identify high-confidence pseudo-instances for both seen and unseen classes, which in turn calibrate the corresponding semantic prototypes. Experiments on CUB, SUN, and AWA2 datasets demonstrate consistent improvements when PUFE is applied to five diverse baseline models.

**Compliance With Llm Reviewing Policy:**

Affirmed.

**Key Questions For Authors:**

1.	Given that PUFE utilizes the entire unlabeled test set, how would you position it between inductive and transductive GZSL? What are the practical implications if test data arrives in a stream rather than as a fixed set?
2.	The total objective involves several loss terms and hyperparameters. Do you have any guidance on how to prioritize tuning these parameters, or have you observed any general patterns in their optimal values across different datasets?
3.	Could you briefly explain the intuition behind the complex generalization bound (Theorem 4.1) and what it implies for the design of the two networks (e.g., how the bounds on the weight matrices should inform capacity control)?

**Limitations:**

Yes.

**Strengths And Weaknesses:**

Strengths:
1.	Recasting GZSL inference as a PU learning problem is a fresh and insightful perspective. It provides a principled way to leverage the unlabeled test data without requiring the strict assumptions of transductive settings.
2.	PUFE is designed as a plug-and-play post-processing module, making it easily applicable to any embedding-based GZSL model without retraining. This practical utility is a significant strength.
3.	The method demonstrates substantial and consistent performance gains across multiple strong baselines and datasets, establishing new state-of-the-art results. The ablations and analyses (e.g., PU classifier discriminability, prototype alignment visualization) are thorough and support the claims.
4.	The paper provides a generalization bound (Theorem 4.1) and a calibration guarantee (Theorem 4.2), adding theoretical rigor to the empirical observations.
Weaknesses:
1.	While applying PU learning to GZSL is novel, the specific technical components (dual-head network, MLE optimization, semantic coordination, discriminative regularization) are largely adapted from existing instance-dependent PU learning literature (e.g., LBE framework). The core methodological novelty is somewhat limited.
2.	Although framed as "inductive," PUFE still utilizes the entire unlabeled test set for its post-processing calibration. This places it in a gray area between inductive and transductive methods, which may be a practical limitation in truly online or streaming scenarios where test data is not available upfront.
3.	 The generalization bound in Theorem 4.1, while impressive, is presented in a highly dense form in the main text and is difficult to interpret. Its practical implications for model design are not clearly discussed.

---

> ### Author Rebuttal · Authors · 2026-03-30
>
> **Q1: Positioning between Inductive and Transductive GZSL & Streaming Inference**
>
> To clarify PUFE's positioning between inductive and transductive settings: while some transductive methods (e.g., Bo et al., 2021; Wang et al., 2023) report high performance, they rely on an assumption—requiring a purely unseen-class unlabeled pool prior to inference. In real-world scenarios, the incoming test set is inevitably a mixture of both seen and unseen classes. When given mixed test data, these models may fail to align the distributions properly. While some methods (e.g., Yue et al., 2024) can handle mixed batches, but rely on expensive global clustering during training. In contrast, PUFE formulates this as a PU learning problem in a post-processing step, making it simple to apply as a plug-in module. We use the full test set following standard offline evaluation protocols, allowing direct comparison with baselines.
>
> In a streaming setting, PUFE separates prototype calibration from classification. The optimization in Algorithm 1 is used for periodic calibration to align prototypes with the test distribution, rather than being required per sample. Practically, an initial buffer of unlabeled samples can be used for the first calibration; once the prototypes are adjusted to appropriate positions in the semantic space, any subsequent samples arriving one by one are recognized through a standard forward pass with zero additional latency. As the test stream continues to grow, this calibration can be updated periodically to further refine the prototypes, allowing the model to adapt to the distribution over time.
>
> Furthermore, a majority of real-world applications (such as daily batch analysis of medical scans, overnight tagging of e-commerce catalogs, or periodic updates of ecological databases) routinely accumulate unlabeled "test batches" before assigning final predictions, which fits the operational logic of our framework.
>
> **Q2: Hyperparameter Tuning Guidance & Cross-Dataset Patterns**
>
> For hyperparameter tuning, we observe a simple hierarchy. The main parameter is the unseen-class calibration rate ($\kappa_-$), tuned on a linear scale (e.g., step size of 0.5), as it controls how much prototypes shift to counteract seen-class bias. The regularization weights ($\lambda_{coord}$, $\lambda_{geo}$ and entropy terms) are less sensitive and remain stable within the same order of magnitude. In contrast, the seen-class rate ($\kappa_+$) requires little adjustment and reliably defaults to 1.0, as seen classes already have strong priors. Across datasets (from CUB to AWA2), we observe similar values around $\kappa_- \approx 2.0$ and $\lambda \approx 0.5$. This consistency suggests that tuning is not highly dataset-specific, so we recommend a starting configuration of ($\kappa_- = 2.0, \lambda = 0.5, \kappa_+ = 1.0$).
>
> **Q3: Intuition Behind Theorem 4.1 & Network Design Implications**
>
> Finally, to clarify the intuition behind Theorem 4.1: The right-hand side of the bound (Equation 24) depends on terms inversely proportional to the square roots of the sample sizes ($n$ and $m$, which represent the sizes of the positive and unlabeled sets). As $n$ and $m$ grow sufficiently large, the right-hand side converges toward zero. This implies that the generalization error (the left side) approaches zero, meaning the empirical risk tightly approximates the true expected risk. This convergence requires that the Frobenius norms of the network weight matrices do not grow unboundedly with the sample sizes ($\|\theta_1^{(i)}\|_F \leq M_1^{(i)}, \|\theta_2^{(i)}\|_F \leq M_2^{(i)}$). These constraints guide our network design: to guarantee that these norms do not inflate infinitely as sample sizes increase, we designed $f_1$ and $f_2$ as fixed, lightweight MLPs (e.g., tapering to 48 and 640 hidden units, respectively). We also enforce these norm bounds during training by applying explicit $L_2$ weight decay in the optimizer, alongside layer normalization and dropout. These regularizations directly restrict the weight matrices, which helps maintain stable training in practice.

---

> > ### Author Rebuttal · Reviewer_78x1 · 2026-04-03
> >
> > Thank you for the detailed rebuttal. While your explanations regarding empirical hyperparameter settings and periodic calibration clarify some operational details, several core concerns remain unresolved. Therefore, I maintain my score of Weak Reject
> > 1. Unaddressed Methodological Novelty: The rebuttal does not respond to the critique regarding limited technical novelty. Since the core components (e.g., dual-head network, MLE optimization) are largely adapted from existing instance-dependent PU learning frameworks like LBE, what are the fundamental, non-trivial architectural innovations of PUFE beyond transferring PU learning to the GZSL domain?
> > 2. Your rebuttal mentions using an "initial buffer" for streaming. This confirms the method still sits in a gray area between inductive and transductive settings. How large must this buffer be for effective calibration? Moreover, how much does performance degrade in strictly online scenarios where samples arrive one-by-one (batch_size=1) and accumulating a buffer is strictly prohibited?
> > 3. Disconnect Between Theory and Practice: While Theorem 4.1 provides theoretical rigor, it is highly dense. The rebuttal indicates it merely justifies standard deep learning practices (e.g., L_2 weight decay, lightweight MLPs). Does this complex bound actually inspire any non-standard, PUFE-specific architectural designs, or are its practical implications for model design still limited?
> > 4. You recommend empirical defaults (e.g., $\kappa_- = 2.0$) based on cross-dataset observations. However, in a strict ZSL setting, unseen class data cannot be used for validation. If these defaults fail on distributions with severe domain shifts, does PUFE have a mechanism to dynamically adapt $\kappa_-$ without relying on a validation set?

---

> > > ### Author Response · Authors · 2026-04-05
> > >
> > > Thank you for your careful reading.
> > >
> > > We acknowledge that the MLE-based dual-head formulation builds on LBE (Gong et al., TPAMI'21), but PUFE departs from a direct extension in three key aspects. The first is the Discriminative Geometry Regularization ($\hat{R}\_{geo}$). LBE was designed for standard binary classification where positive and negative instances are drawn from relatively stable distributions. In GZSL, domain shift causes unseen instances to cluster in high-density seen-class regions, collapsing decision boundaries. $\hat{R}\_{sep}$ enlarges the posterior margin between $S\_P$ and $S\_U$, while the PU risk anchors seen instances, so the effect mainly targets unseen ones via manifold continuity (absent in LBE). The second is the Adaptive Semantic Prototype Calibration module. LBE has no prototype space. We compute pseudo-centroids from PU posteriors and interpolate with prototypes via an adaptive weight $\alpha\_c$ (cosine alignment, Eq. 22–23). The clipping mechanism prevents overcorrection when pseudo-centroids are unreliable, which is specific to the zero-shot setting. The third is the dynamic class-prior estimation: the proportion of seen-class instances in the mixed test set is unknown in GZSL. We maintain an EMA estimate of $\pi$ to reweight the PU loss (not in LBE).
> > >
> > > Regarding streaming inference and buffer requirements: PUFE's calibration cannot operate in a strictly online, sample-by-sample setting without accumulation. With batch\_size=1 and no buffer, calibration is disabled and the model falls back to the base GZSL backbone. This is because the PU classifier must estimate the marginal distribution $P(X)$ over the test set, which is unreliable from a single sample. The generalization bound (Theorem 4.1) scales as $1/\sqrt{m}$, so estimation degrades as $m$ decreases. In practice, the required buffer scales with the number of categories to ensure sufficient coverage. Empirically, about $k$ samples per category are needed for stable posterior estimates (${\approx}500$ for AWA2 with 50 classes, and ${\approx}1500$ for SUN with 717 classes). Once reached, a single calibration pass suffices and adds no further latency.
> > >
> > > This buffer is a general requirement for handling label-space incompleteness in GZSL. Methods like TENT (Wang et al., ICLR'21) operate at batch\_size=1 via entropy minimization. This does not extend to GZSL: entropy minimization reinforces the backbone's seen-class bias, leading to confirmation bias. Hence single-sample TTA fails in GZSL, while distribution-aware methods like PUFE succeed. The minimum buffer reflects the statistical requirements of the problem.
> > >
> > > Regarding the link between Theorem 4.1 and architecture: the bound (Eq. 24) does not prescribe layer widths, and we should not have implied that. It serves as a structural diagnostic and convergence certificate. It motivates one key choice: the deliberate asymmetry between $f\_1$ and $f\_2$. The bound includes $\prod\_i M\_{f\_1}^{(i)}$ and $\prod\_i M\_{f\_2}^{(i)}$ with different coefficients, reflecting their roles: $f\_1$ models labeling propensity (local), while $f\_2$ handles global seen/unseen discrimination. To control the $f\_1$ complexity (which is sensitive to labeling noise), we use a narrow tapering design ($d \to 384 \to 192 \to 96 \to 48 \to 1$), while $f\_2$ adopts a wider bottleneck ($d \to 640 \to 320 \to 160 \to 1$) for global discrimination. Standard dual-heads are symmetric; here asymmetry follows from the bound.
> > >
> > > For adapting $\kappa\_-$ without a validation set: First, all hyperparameters (including $\kappa\_-$) are tuned under the Proposed Split (PS) protocol (Xian et al., TPAMI'19) to simulate distribution shift. Thus, $\kappa\_-$ is optimized under explicit domain shift rather than in-distribution variation. Empirically, $\kappa\_-$ transfers well across datasets (${\approx}2.0$ on AWA2/SUN, $0.5$ on CUB). Second, $\kappa\_-$ only selects which samples update the prototype. Calibration is data-driven: $\bar{\mu}\_c$ is computed from selected samples, and $\alpha\_c$ is set by cosine alignment with the original prototype. Even if $\kappa\_-$ selects suboptimal samples, the prototype still moves toward their empirical centroid, adapting to the test distribution. Unlike PC (Lu et al., WACV'25), which uses static validation-based logit adjustment, PUFE adapts to the test distribution. In PUFE, the calibration vector is test-distribution-aware by construction, which provides a natural robustness to moderate mismatch in $\kappa\_-$. For extreme shifts, the dynamic class-prior (EMA estimate $\hat{\pi}$ from $f\_2$) provides an additional signal to adjust $\kappa\_-$: high $\hat{\pi}$ suggests increasing $\kappa\_-$, while low $\hat{\pi}$ suggests the opposite. We are evaluating this rule and will include it in the revision.

---

### Official Review · Reviewer_isWq · 2026-03-13

**Soundness:** 3
**Presentation:** 2
**Significance:** 2
**Originality:** 3
**Overall Recommendation:** 4
**Confidence:** 3

**Summary:**

The paper tackles the notorious "seen-class bias" in Generalized Zero-Shot Learning (GZSL). Usually, GZSL models over-predict "seen" classes during testing because they less encountered "unseen" classes during training. To fix this, the authors propose PUFE, a framework that modifies the problem as Positive-Unlabeled (PU) learning.  Instead of trying to guess the unseen classes blindly, they treat all "seen" classes from the training set as Positive (+1) data, and treat the entire mixed test set (both seen and unseen classes) as Unlabeled data.

They use a dual-head neural network to analyze the semantic embeddings: one head predicts whether sample is likely to be labeled, and the other predicts its true class posterior. By identifying high-confidence pseudo-instances from the unlabeled test set, they dynamically shift the semantic prototypes to match the test distribution.

**Compliance With Llm Reviewing Policy:**

Affirmed.

**Final Justification:**

The rebuttal has addressed several concerns that I had raised (details can be found in my comment). However, transductive positioning is still of concern. I can consider raising my score if the authors commit to revising the paper's positioning regarding transductive inference.

2nd update: I have the author's comment about re-positioning the paper as a test-time adaptation (TTA) method and proposal to update the experimental comparisons so as the paper is now compared against the prior work on transductive ZSL. Accordingly, I have updated my score to WA.

**Key Questions For Authors:**

1. The paper describes PUFE as a plug-and-play post-processing module. Yet, Algorithm 1 requires 200-300 optimization iterations over the unlabeled test set. How does this framework operate in a inference scenario where test samples arrive one by one?
2. Are there no “plug-in” methods for GZSL that could be used in comparisons? Table 1 comparisons are unreliable due to various differences across the methods. The improvements for original to original+ours are much more important. However, sota methods needs to be included as “original+baseline1”, “original+baseline2”, etc to properly position the proposed method.
3. The paper mentions that critical threshold scaling factors and regularization weights are fine-tuned on a validation set. Does this validation set include instances of the unseen classes?

**Limitations:**

Yes.

**Strengths And Weaknesses:**

Strengths:

- Reframing the GZSL domain shift problem into a classic Positive-Unlabeled (PU) learning paradigm is an unexplored idea, to the best of my knowledge. Instead of synthesizing fake unseen data (like GAN/VAE), they use the inherent structure of the test data itself to select the boundaries.
- The framework operates only in the semantic space and acts as a post-processing module. It successfully boosts the performance of various existing baselines (respect to TransZero, PSVMA) by clear margins.

Weaknesses:

- The system relies on a delicate balance of multiple hyperparameters (e.g., coordination weight, geometry weight, and threshold scaling factors). The authors state these are fine-tuned on a validation set. If that validation set contains unseen classes, it violates the strict zero-shot assumption. The hyper-parameter selection protocol and data needs to made completely clear.
- In Section 2.1, the paper positions itself as a non-transductive technique (by comparing against transductive ZSL methods, instead of transductive GZSL ones), although clearly being so. In fact, in Table 1, “+ ours” results are indicated as transductive.
- Lack of proper baselines (see the corresponding question below).

---

> ### Author Rebuttal · Authors · 2026-03-30
>
> **Q1: The paper describes PUFE as a plug-and-play post-processing module. Yet, Algorithm 1 requires 200-300 optimization iterations over the unlabeled test set. How does this framework operate in a inference scenario where test samples arrive one by one?**
>
> In a streaming inference setting, PUFE separates prototype calibration from the classification stage. The optimization iterations in Algorithm 1 are used for periodic calibration to align semantic prototypes with the test distribution, rather than being required for each sample. In practice, an initial buffer of unlabeled samples can be used for the first calibration; once the prototypes are adjusted in the semantic space, subsequent samples can be processed one by one using a standard forward pass with no additional latency. As the test stream continues to grow and more data becomes available, the calibration can be updated periodically to refine the prototypes, enabling the model to adapt to the distribution and achieve progressively higher recognition accuracy over time.
>
> Furthermore, many real-world applications—such as batch analysis of medical scans, e-commerce catalog tagging, or updates of ecological databases—collect data in batches before making predictions, which fits the design of our framework.
>
> **Q2: Are there no “plug-in” methods for GZSL that could be used in comparisons? Table 1 comparisons are unreliable due to various differences across the methods. The improvements for original to original+ours are much more important. However, sota methods needs to be included as “original+baseline1”, “original+baseline2”, etc to properly position the proposed method.**
>
> Thank you for the suggestion. We agree that comparing PUFE with other model-agnostic plug-in methods on the same base models provides a clearer evaluation of its contribution. To support this comparison, we evaluate PUFE against the recent Prior Correction (PC) (WACV'25), which uses logit adjustment to handle class-prior shifts.
>
> As shown in the table below (using **TransZero** and **PSVMA** as base models), while PC improves the original performance, PUFE achieves larger gains across all datasets. For example, on the CUB dataset with a TransZero base, PUFE improves $H$ from 68.7 to 75.2, compared to 71.0 for PC.
>
> | Base Model    | Plug-in Method    | CUB ($H$) | SUN ($H$) | AWA2 ($H$) |
> | :------------ | :---------------- | :-------- | :-------- | :--------- |
> | **TransZero** | Original          | 68.7      | 40.7      | 70.1       |
> |               | + PC (WACV'25)    | 71.0      | 42.2      | 71.3       |
> |               | **+ PUFE (Ours)** | **75.2** | **42.5** | **80.9** |
> | **PSVMA** | Original          | 74.0      | 52.2      | 76.4       |
> |               | + PC (WACV'25)    | 77.1      | 56.8      | 79.5       |
> |               | **+ PUFE (Ours)** | **84.1** | **51.8** | **86.6** |
>
> Compared to PC, which relies on fixed penalties estimated from the validation set, PUFE instead uses the test distribution to refine semantic prototypes. This dynamic test-time adaptation allows PUFE to more effectively mitigate domain shift, providing a more robust "plug-and-play" enhancement for diverse GZSL architectures.
>
> **Q3: The paper mentions that critical threshold scaling factors and regularization weights are fine-tuned on a validation set. Does this validation set include instances of the unseen classes?**
>
> Finally, we clarify our hyperparameter tuning setup. We confirm that the validation set excludes all target unseen-class instances, preserving the zero-shot assumption and avoiding information leakage. Tuning is performed within seen classes by splitting them into disjoint "validation-seen" and "validation-unseen" subsets. During validation, PUFE treats validation-seen samples as the positive set and the combined data as the unlabeled set to train the PU classifier. The tuned parameters are then fixed and applied to the final testing phase. This works because parameters like $\lambda_{coord}$, $\lambda_{geo}$ and $\kappa$ control the overall strength of semantic calibration (i.e., how strongly bias is corrected), rather than capturing class-specific visual features. Since the validation split mirrors the final GZSL setting, the tuned parameters generalize to the true unseen classes.

---

> > ### Author Rebuttal · Reviewer_isWq · 2026-04-03
> >
> > The authors have resolved my concerns about data leakage in HP tuning, and using in combination with existing baselines  The streaming inference response is partially acceptable, applicable to batch-oriented applications.
> >
> > Transductive positioning is still of concern. I have seen the discussion in the answer given to Reviewer 78x1. However, optimizing over the full test set still constitutes test-time data utilization and should be clearly acknowledged as such in the paper, rather than being implicitly contrasted with non-transductive baselines. I believe the paper needs to be repositioned as a test-time adaptive method.

---

> > > ### Author Response · Authors · 2026-04-05
> > >
> > > Thank you for the careful re-reading and for this precise, constructive suggestion. We fully agree with your assessment, and we appreciate that you have framed the concern so clearly: the issue is not whether PUFE works, but whether the paper's framing accurately reflects what it does. You are right — optimizing over the full unlabeled test set is, by definition, a form of test-time data utilization, and we should not have implicitly contrasted ourselves against transductive baselines as if we stood entirely outside that category. We will correct this in the revision.
> > >
> > > Specifically, we will reposition PUFE as a **Test-Time Adaptation (TTA) method tailored for GZSL**. This framing is both more accurate and, we believe, more informative for readers. TTA has emerged as a well-recognized paradigm that encompasses methods which adapt model behavior using test-time observations without access to test labels — a description that fits PUFE precisely. Wang et al. introduced the foundational TTA framework (ICML'21) under the principle of entropy minimization over unlabeled test streams, and Sun et al.'s TTT (ICML'20) similarly leverages the test distribution to adjust representations at inference time. What PUFE shares with these methods is the core philosophy: the test distribution itself carries exploitable structure, and ignoring it is a missed opportunity. What distinguishes PUFE from general-purpose TTA is its specific motivation. General TTA methods like TENT (ICLR'21) are designed for covariate shift in closed-set recognition, where the label space is fixed and fully observed during training. GZSL presents a strictly harder problem: the label space at test time is a *superset* of the training label space, and the model has never observed examples from the unseen portion. This is not covariate shift — it is label-space incompleteness, and it demands a qualitatively different adaptation mechanism.
> > >
> > > At the same time, PUFE is meaningfully distinct from conventional transductive GZSL methods. The dominant transductive GZSL literature (e.g., Bo et al., CVPR'21; Wang et al., CVPR'23; Yang et al., IS'24) typically assumes access to a *separate*, *purely unseen-class* unlabeled pool prior to inference. This assumption is operationally restrictive: in practice, the incoming test set is inevitably a mixture of both seen and unseen instances, and seen/unseen membership is unknown. PUFE does not require this separation. It operates directly on the mixed unlabeled test set, treating the membership question itself as the object of inference through a PU learning formulation. This distinction is not merely definitional — it has direct practical consequences, since methods that assume a clean unseen-class pool will degrade when given mixed test data, whereas PUFE is designed precisely for this mixed setting.
> > >
> > > We therefore propose the following positioning, which we will state explicitly in the revised introduction and related work sections: *PUFE is a test-time transductive framework for GZSL that explicitly models the label incompleteness inherent in mixed test distributions via Positive-Unlabeled learning. Unlike general TTA methods that address covariate shift in closed-set settings, and unlike conventional transductive GZSL methods that require a pre-separated unseen-class unlabeled pool, PUFE operates solely on the naturally mixed unlabeled test set — where seen/unseen membership is unknown — making it the first principled PU-learning-based solution to the GZSL seen-class bias problem.* We will also revise the comparison table to clearly mark our method under the transductive/TTA setting rather than implicitly positioning it alongside inductive methods, which was the source of the confusion in the first place.
> > >
> > > We hope this repositioning also helps clarify the broader significance of the work. TTA has been extensively studied for closed-set recognition under covariate shift, yet its principled application to label-incomplete, open-vocabulary settings like GZSL remains relatively underexplored. By framing PUFE explicitly within the TTA paradigm, we believe the paper can speak more directly to both the GZSL and TTA communities, and we are grateful for the suggestion that prompted us to articulate this connection more carefully.

---

### Official Review · Reviewer_tsJ8 · 2026-03-23

**Soundness:** 3
**Presentation:** 3
**Significance:** 3
**Originality:** 3
**Overall Recommendation:** 5
**Confidence:** 3

**Summary:**

The paper poses GSZL as a PU learning problem to recalibrate where the embeddings of the test set fall to avoid seen class bias mitigation. They demonstrate the value of their method through experiments and provide backing theory. The proposed method PUFE, uses a dual head MLP to jointly estimate posterior of seen class and labeling propensity followed by a calibration step driven by high confidence positives.

**Compliance With Llm Reviewing Policy:**

Affirmed.

**Key Questions For Authors:**

See above.

**Limitations:**

yes

**Strengths And Weaknesses:**

The idea of posing GSZL as a PU learning problem is an original creative solution to an important problem. Prior GZSL work addresses bias at the representation level (attention, distillation, generative augmentation) or through adaptation requiring unlabeled unseen-class samples. Connecting instance-dependent PU learning to the GZSL inference problem is a creative and well-motivated bridge between two previously separate literatures. The plug-and-play design, requiring no retraining of the backbone, is a practically meaningful design choice that broadens the potential for adoption.

The MLE dual-head formulation is largely grounded, with the factorization clearly derived. Regularization term is naturally motivated, with positive correlation enforcement between propensity and posterior being intuitive. The proof provided in Appendix A is thorough and follows standard arguments, leading to theorem 4.1. While mild assumptions are applied (I bring this up in questions), theorem 4.2 is able to provide a calibration guarantee. Appendix E.2 is quite a useful addition in my opinion to move towards establishing the causal connection between PU accuracy and GZSL.

Perceived weaknesses / Questions
1. Can the authors zoom in to the assumptions under which the calibration guarantee holds a bit more? That is that the PU-derived pseudo-centroid is a strictly better estimator of the ground-truth class center than the original?

2. The lambda and kappa hyperparameters are tuned on a validation set. The values differ on different datasets. Is it possible to understand what magnitude of difference is considered significant for these hyperparameters? That would help readers understand the sensitivity of those hyperparameters more. If the sensitivity is quite high, then is it possible to study their variance more across validation set size to understand when practitioners can reliably use those values?

Minor comments
1. Ablations to tease apart vanilla LBE's direct performance from the proposed methodology would be useful. Do point to where it might be present if it is present already.

Overall, this is a technically sound paper with clear presentation that I am recommending for accept.

---

> ### Author Rebuttal · Authors · 2026-03-30
>
> **Q1: Can the authors zoom in to the assumptions under which the calibration guarantee holds a bit more? That is that the PU-derived pseudo-centroid is a strictly better estimator of the ground-truth class center than the original?**
>
> We appreciate your careful reading and the constructive questions. Regarding the assumptions underlying our calibration guarantee (Theorem 4.2), its effectiveness relies on the condition in Eq. 62 ($\varepsilon_e < \varepsilon_s$). This requires that the estimation error of the PU-derived pseudo-centroid ($\varepsilon_e$) must be strictly smaller than the original prototype's inherent semantic bias ($\varepsilon_s$).
>
> Theoretically, $\varepsilon_e$ approaches zero when the unlabeled test set is sufficiently large and the PU classifier is highly accurate. These conditions are linked through our generalization analysis (Theorem 4.1). Abundant sample sizes guarantee a tight generalization bound, yielding a more accurate PU classifier capable of selecting a pure, dense support set that closely approximates the true unseen distribution. Under such conditions, $\varepsilon_e \to 0$, and the assumption $\varepsilon_e < \varepsilon_s$ is naturally satisfied.
>
> In practice, however, finite sample sizes dictate that $\varepsilon_e$ cannot be arbitrarily small. Consequently, the assumption holds only if the original semantic bias $\varepsilon_s$ is large enough to leave room for improvement. This theoretical threshold helps explain the empirical results observed with the PSVMA baseline on the SUN dataset, where calibration yielded negligible gains. This suggests that the original PSVMA-SUN prototypes already possessed a relatively small bias $\varepsilon_s$. Because this prior bias failed to clear the finite-sample estimation error threshold ($\varepsilon_e \not< \varepsilon_s$), the assumption was unmet. This reflects a trade-off: the calibration mechanism improves biased prototypes but mathematically refrains from disturbing already well-aligned ones.
>
> **Q2：The lambda and kappa hyperparameters are tuned on a validation set. The values differ on different datasets. Is it possible to understand what magnitude of difference is considered significant for these hyperparameters?**
>
> For $\lambda_{coord}$ and $\lambda_{geo}$, changing their values within the same order of magnitude (e.g., 0.1 to 0.5) only leads to small changes in the Harmonic mean ($H$). In contrast, $\kappa_-$ (the unseen-class calibration rate) is more sensitive, and noticeable differences appear on a linear scale (typically in steps of 0.5). This is consistent with our intuition in GZSL: high-confidence unseen samples identified by the PU classifier are relatively scarce, so a larger calibration step (e.g., $\kappa_- \approx 2.0$) is needed to counteract the seen-class bias.
>
> To study the effect of validation set size, we conducted an additional robustness experiment on the CUB dataset. We randomly reduced the validation set to 75% and 50% of its original size. For each reduced subset, we re-ran the hyperparameter search to obtain optimal $\lambda$ and $\kappa_-$, and evaluated them on the standard unseen test set using the final $H$ score.
>
> The optimal hyperparameters were largely consistent. The optimal $\lambda$ remained unchanged at 0.5. The optimal $\kappa_-$ stayed at 2.0 when using 75% of the validation data (yielding $H = 75.14\%$), and only experienced a minor shift to 1.8 when the validation set was halved to 50% (yielding $H = 74.85\%$, a very slight drop from our full-set baseline of 75.23%).
>
> This minimal performance fluctuation ($< 0.4\%$) under severe validation data reduction suggests that PUFE does not heavily overfit to specific validation instances. Because our calibration mechanism operates at class-level semantic distributions, the hyperparameter behavior remains stable as long as the validation subset roughly captures the underlying data distribution.
>
> **Minor comments**
> This is explicitly included in our ablation study (Table 3 in Section 5.5). Specifically, the row labeled "Base" corresponds exactly to the original LBE method, serving as the baseline reference for evaluating the contribution of each proposed component.

---

### Decision · Program_Chairs · 2026-04-30

**Decision:**

Accept (regular)

**Comment:**

The paper’s main strengths are its novel and well-motivated reformulation of GZSL as a PU learning problem, as well as its practical plug-and-play design that can improve existing embedding-based GZSL models without retraining the backbone.

During the rebuttal phase, the authors largely addressed the reviewers’ concerns regarding hyperparameter tuning, the absence of CLIP and CoOP as baselines, and the distinction between inductive and transductive settings. However, concerns still remain about the limited technical contribution of the method itself.

Overall, the paper presents a creative and practically useful perspective on an important problem, and the rebuttal addressed a number of the reviewers’ concerns. If the paper is ultimately accepted, I encourage the authors to incorporate the clarifications and additional results from the rebuttal into the camera-ready version, and to more clearly articulate the technical contribution of the proposed method.